# Scavenger receptor endocytosis controls apical membrane morphogenesis in the *Drosophila* airways

**Ana Sofia Pinheiro**[1†], **Vasilios Tsarouhas**[1*†], **Kirsten André Senti**[1,2], **Badrul Arefin**[1,3], **Christos Samakovlis**[1,4*]

[1]Science for Life Laboratory, Department of Molecular Biosciences, The Wenner-Gren Institute, Stockholm University, Stockholm, Sweden; [2]IMBA – Institute of Molecular Biotechnology, Austrian Academy of Sciences, Vienna, Austria; [3]Sahlgrenska Academy, Gothenburg University, Gothenburg, Sweden; [4]Cardiopulmonary Institute, Justus Liebig University of Giessen, Giessen, Germany

**Abstract** The acquisition of distinct branch sizes and shapes is a central aspect in tubular organ morphogenesis and function. In the *Drosophila* airway tree, the interplay of apical extracellular matrix (ECM) components with the underlying membrane and cytoskeleton controls tube elongation, but the link between ECM composition with apical membrane morphogenesis and tube size regulation is elusive. Here, we characterized Emp (epithelial membrane protein), a *Drosophila* CD36 homolog belonging to the scavenger receptor class B protein family. *emp* mutant embryos fail to internalize the luminal chitin deacetylases Serp and Verm at the final stages of airway maturation and die at hatching with liquid filled airways. Emp localizes in apical epithelial membranes and shows cargo selectivity for LDLr-domain containing proteins. *emp* mutants also display over elongated tracheal tubes with increased levels of the apical proteins Crb, DE-cad, and phosphorylated Src (p-Src). We show that Emp associates with and organizes the βH-Spectrin cytoskeleton and is itself confined by apical F-actin bundles. Overexpression or loss of its cargo protein Serp lead to abnormal apical accumulations of Emp and perturbations in p-Src levels. We propose that during morphogenesis, Emp senses and responds to luminal cargo levels by initiating apical membrane endocytosis along the longitudinal tube axis and thereby restricts airway elongation.

## Editor's evaluation

In this important work, the authors convincingly show that the *Drosophila* scavenger receptor Emp (homologous to human CD36) senses and responds to the levels of its cargo apical ECM proteins and triggers the initiation of apical endocytosis, thereby regulating tube length via controlling Crumbs and Src. This work will be of broad interest to cell and development biologists as well as cancer biologists.

## Introduction

The tube shapes in transporting organs like kidney, lung, and vascular system are precisely controlled to ensure optimal fluid flow and thereby, function. Failure in normal tube size acquisition leads to cystic, stenotic, or winding tubes. The *Drosophila* respiratory network, the trachea, provides a well-characterized system for the genetic dissection of tubular organ maturation. Like mammalian lungs, the trachea undergoes a precisely timed series of maturation events to convert the nascent branches into functional airways. First, a transient secretion burst of luminal proteins

**\*For correspondence:**
Vasilios.Tsarouhas@su.se (VT);
christos.samakovlis@scilifelab.se (CS)

[†]These authors contributed equally to this work

**Competing interest:** The authors declare that no competing interests exist.

(*Tsarouhas et al., 2007*; *Jayaram et al., 2008*; *Förster et al., 2010*) initiates diametric tube expansion. Luminal proteins assemble into a chitinous central rod and into a cuticular apical extracellular matrix (aECM, taenidia), lining the apical membrane. After 10 hr, luminal material becomes rapidly cleared from the tubes by massive endocytosis involving several endocytic pathways. Finally, a liquid clearance pulse converts the tubes into functional airways (*Tsarouhas et al., 2007*). Genetic studies suggested an instructive role of luminal chitin and proteins in tube growth coordination and termination. Mutations affecting chitin biosynthesis (*kkv*) or matrix assembly (*knk*, *gasp*) show irregular tube shapes, diametric expansion and tube maturation defects (*Moussian et al., 2006*; *Tiklová et al., 2013*; *Öztürk-Çolak et al., 2016*). Tube elongation is continuous during tracheal development and its termination requires chitin biosynthesis and the secreted chitin deacetylases vermiform (Verm) and serpentine (Serp) (*Beitel and Krasnow, 2000*; *Luschnig et al., 2006*; *Wang et al., 2006*; *Öztürk-Çolak et al., 2016*). These luminal proteins presumably modify the structure and physical properties of the aECM and thereby restrict tube elongation. In addition to the luminal matrix pathway, components involved in the assembly of basolateral septate junctions (SJs) also restrict tube elongation, through the regulation of the subcellular localization of Crumbs (Crb), a transmembrane protein that promotes expansion of the tracheal cell apical surface and tube elongation (*Laprise et al., 2010*). More recently, the conserved non-receptor tyrosine kinase, Src oncogene at 42A (Src42A) was found to promote axial elongation by controlling the apical cytoskeleton and apical cell (*Förster and Luschnig, 2012*; *Nelson et al., 2012*; *Öztürk-Çolak et al., 2016*; *Olivares-Castiñeira and Llimargas, 2018*). Additionally, Yorkie (Yki) and several components of the Hippo pathway control tube elongation, along with transcription factors like Blimp-1 and Grainy head (Grh) (*Hemphälä et al., 2003*; *Robbins et al., 2014*; *Öztürk-Çolak et al., 2016*; *McSharry and Beitel, 2019*; *Skouloudaki et al., 2019*). An appealing model suggests that the interaction between apical membrane and aECM elasticity may influence apical cytoskeletal organization and thereby control tube shapes (*Dong et al., 2014*). Although ECM integrity and the apical cytoskeleton appear crucial in tube length regulation, it is unknown how ECM signals are perceived by the airway cells to regulate their shapes during tube maturation.

Scavenger receptors comprise a superfamily of cell surface membrane proteins that bind and internalize modified lipoproteins and various other types of ligands. Cluster of differentiation 36 (CD36) belongs to class B scavenger receptor family, which includes scavenger receptor B1 (SRB1) and lysosomal integral membrane protein 2 (LIMP2). CD36 is expressed on the surface of many cell types including epithelial, endothelial cells, and macrophages. Disruption of CD36 function in mice can lead to inflammation, atherosclerosis, metabolic disorders, tumor growth, and metastasis (*Chen et al., 2008*; *Pascual et al., 2017*; *Wang et al., 2020*). CD36 has several cargoes, including long-chain fatty acids, oxidized LDL (ox-LDL), oxidized phospholipids and thrombospondin-1 (TSP-1) (*Githaka et al., 2016*; *Deng et al., 2022*). In vitro imaging studies of macrophages and endothelial cells propose that CD36 clustering at the cell surface upon engagement of multivalent ligands and in conjunction with the cortical cytoskeleton triggers signal transduction and receptor–ligand complex endocytosis (*Jaqaman et al., 2011*; *Githaka et al., 2016*). The activity of several signaling effectors, including the Src family kinases, Fyn, Yes (*Thorne et al., 2006*; *Zani et al., 2015*) and the mitogen-activated kinases, Jun-kinase (JNK) 1 and 2 (*Rahaman et al., 2006*) can be regulated by CD36. The *Drosophila* genome includes a family of 14 CD36-like genes, with distinct tissue-specific expression patterns. The genetic analysis of a few members in this class B scavenger receptor family implicated them in phagocytosis, immune responses, and photoreceptor function (*Philips et al., 2005*; *Stuart et al., 2005*; *Voolstra et al., 2006*). The *Drosophila* epithelial membrane protein (Emp) shows the highest similarity with CD36 and is selectively expressed in embryonic epithelial tissues (*Hart and Wilcox, 1993*).

Here, we show that Emp is a selective receptor for internalization, endosomal targeting, and tracheal luminal clearance of proteins with LDLr-domains. *emp* mutants display over elongated tracheal tubes with increased levels of junctional Crb, DE-cad, and phospho-Src. Reduction of Src42A in *emp* mutants, rescues the tube elongation phenotype indicating that Emp modulates junctional *p*-Src42A levels to control apical membrane expansion and tube length. The organization of the beta-heavy spectrin (βH-Spectrin) cytoskeletal network is compromised in *emp* mutants. Emp binds to βH-Spectrin directly, suggesting that it provides a direct link between ECM, apical membrane, and cytoskeleton during tube maturation process. Re-expression of human CD36 in *emp* mutants can ameliorate the mutant tube phenotypes suggesting conserved functions of Emp.

## Results

**Emp is a selective scavenger receptor required for tube elongation and luminal protein clearance**

*emp* (or CG2727 in Flybase) encodes a class-B scavenger receptor expressed in embryonic ecto-dermal epithelial tissues including the tracheal system (*Hart and Wilcox, 1993*). To elucidate the developmental functions of *emp* in the airways, a deletion mutant (*emp^e3d1^*, referred as *emp* mutant hereafter) was generated using the FLP/FRT recombinase system (*Parks et al., 2004*; *Figure 1—figure supplement 1A*). PCR mapping of genomic DNA identified a 4.8-kb deletion encompassing exon 2 in the *CG2727* locus. Both immunofluorescence and western blots using a polyclonal anti-serum against recombinant Emp (see Methods) (*Figure 1—figure supplement 1B*), failed to detect Emp protein in *emp* mutants (*Figure 1—figure supplement 1C, D*). Similarly, quantitative Reverse Transcription-Polymerase Chain Reaction (RT-PCR) of RNA extracted from late embryos showed a strong reduction of *emp* RNA in the mutants (*Figure 1—figure supplement 1E*). *emp* homozygous or *Df(2R)BSC608/emp* mutants were embryonic lethal with few escapers surviving to first instar larvae. This embryonic lethality could be rescued by the re-expression of a transgenic *emp* construct using the ectodermal driver *69BGal4*. This suggests that the deletion generates a strong loss-of function mutation in *emp* and does not affect any neighboring genes required for embryo viability. To examine a potential role of Emp in airway maturation, we visualized the tracheal tubes during embryonic devel-opment in *wild-type* and *emp* mutants. *emp* embryos at stage 16 showed a 30% over-elongation of the dorsal trunk (DT) compared to the *wild-type* (*Figure 1A*), but showed no defects in tube diameter (*Figure 1—figure supplement 1F*). *emp* mutants also failed to fill their airways with gas at hatching (*Figure 1H*, *Figure 1—figure supplement 1I*). Both phenotypes could be rescued by re-expression of *emp* in tracheal cells of the *emp* mutants (*Figure 1A, B, H*, *Figure 1—figure supplement 1Ic*). These data indicate that Emp is required for normal tube elongation and gas-filling during embryonic development. The survival of *emp* mutants overexpressing *emp* in the airways was limited to larval stages, suggesting that the tracheal-specific re-expression of *emp* is not sufficient for larval or adult survival. This suggests additional roles for Emp in other tissues. The human homolog of Emp, CD36 shares the overall protein architecture and 30% of amino acid identity with Emp (*Figure 1—figure supplement 1G, H*). We generated a transgenic line expressing the coding sequence of human CD36 and drove its expression in fly airways to test if Emp and CD36 have conserved functions. We found that both the tracheal length and gas-filling defects in *emp* mutants were partially reversed by tracheal CD36 overexpression, arguing for a conserved function of CD36 (*Figure 1A, B, H*, *Figure 1—figure supplement 1Id*).

To establish whether the failure of tracheal gas-filling originates from a defect in luminal protein clearance, we stained *wild-type* and *emp* mutants for the luminal proteins Serp, Verm, and Gasp. These secretory proteins were internalized from the lumen by late stage 17 in *wild-type* embryos. Serp and Verm, but not Gasp were selectively retained in the *emp* DT airways (*Figure 1C*) suggesting a role of Emp in the internalization of a subset of luminal proteins. To confirm the protein clearance pheno-types, we generated *emp* mutants carrying *btl>Serp-GFP* or *btl>Verm-GFP* or *btl>Gasp-mCherry* transgenes and analyzed them by live imaging from 17 h to 21 h after egg laying (AEL) (*Figure 1D, E*). The three reporters were normally secreted into the lumen of both *emp* mutants and *wild-type* embryos and at 19 hr they were cleared from the tubes of *wild-type* embryos. In *emp* mutants *Gasp-mCherry* was also cleared from the lumen but the Serp-GFP and Verm-GFP reporters remained inside the tube. This suggests that Emp acts as a selective endocytosis receptor during luminal protein clearance. The unperturbed clearance of Gasp-Cherry suggested differential requirements for the internalization of luminal proteins.

*clathrin* (*chc^1^*) null mutants show defects in luminal ANF-GFP clearance (*Tsarouhas et al., 2007*) and tube length (*Behr et al., 2007*). We stained *chc^1^* mutants for Serp, Verm, and Gasp and analyzed them at late embryonic stages, when luminal clearance is completed in *wild-type* embryos. This analysis showed that Serp and Verm were cleared, whereas Gasp clearance was selectively impaired in *chc^1^* mutants (*Figure 1F, G*). This indicates that Emp is involved in a selective, clathrin-independent endo-cytosis pathway internalizing Verm and Serp. Gasp endocytosis is clathrin dependent and presum-ably involves an unidentified surface receptor. These genetic experiments suggest that Emp controls tracheal tube elongation and luminal clearance of chitin deacetylases presumably by mediating their endocytosis.

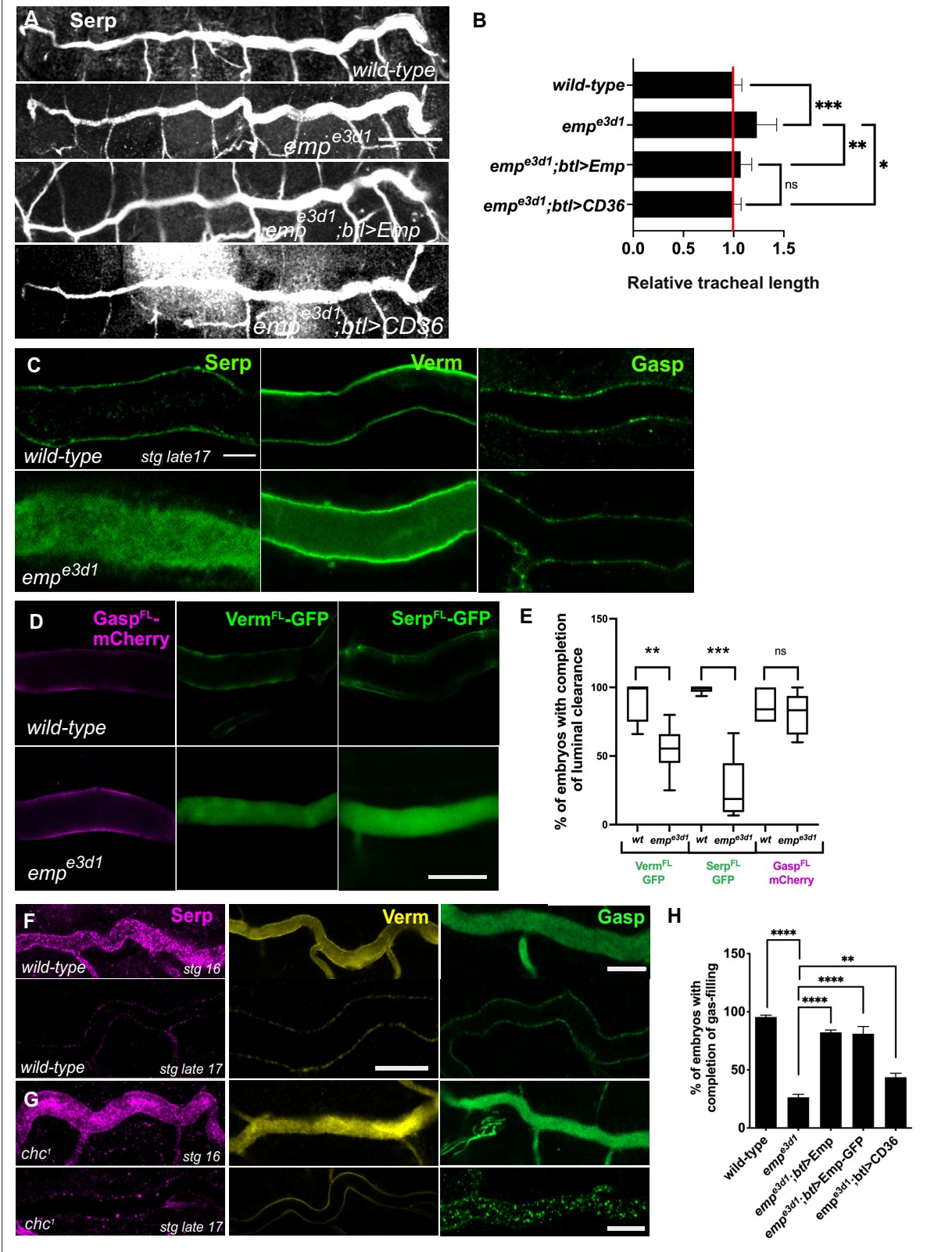

**Figure 1.** *emp^{e3d1}* mutants show over-elongation of the tracheal tubes and severe luminal clearance defect of Serp. (**A**) Images showing the dorsal trunk (DT) of *wild-type*, *emp^{e3d1}*, *emp^{e3d1};btl>Emp*, *emp^{e3d1};btl>CD36* embryos stained for the luminal marker Serp. (**B**) Graph showing the tracheal DT length in *wild-type* embryos (*n* = 16), *emp^{e3d1}* mutants (*n* = 17), *emp^{e3d1};btl>Emp* (*n* = 19) and *emp^{e3d1};btl>CD36* (*n* = 15) embryos. (**C**) Confocal images showing the DT of *wild-type* and *emp^{e3d1}* mutant at late stage 17 embryos, stained for the endogenous luminal proteins Serp, Verm, and

*Figure 1 continued on next page*

*Figure 1 continued*

Gasp. (**D**) Confocal images showing the DT of live *btl>GaspFL-mCherry* (magenta), *btl>VermFL-GFP* and *btl>SerpFL-GFP* (green) in *wild-type*, and *emp$^{e3d1}$* mutant at 20.0 h AEL. (**E**) Plots showing the percentage of embryos with completion of luminal clearance in *btl>VermFL-GFP* (green, *n* = 59), *emp$^{e3d1}$;btl>VermFL-GFP* (green, *n* = 37), *btl>SerpFL-GFP* (green, *n* = 58), *emp$^{e3d1}$;btl>SerpFL-GFP* (green, *n* = 15), *btl>GaspFL-mCherry* (magenta, *n* = 28) and *emp$^{e3d1}$;btl>GaspFL-mCherry* (magenta, *n* = 18). Representative confocal images showing the tracheal DT of *wild-type* (**F**) and *chc$^1$* (**G**) mutant embryos, stained for the endogenous luminal markers Serp (magenta), Verm (yellow), and Gasp (green) before and after luminal clearance, stage 16 and late stage 17 (*n* ≥ 8, for each genotype were analyzed). (**H**) Bar graph shows the percentage of embryos that fill with gas in *wild-type* (*n* = 177), *emp$^{e3d1}$* (*n* = 182), *emp$^{e3d1}$; btl>Emp* (*n* = 119), *emp$^{e3d1}$;btl>Emp-GFP* (*n* = 178), and *emp$^{e3d1}$; btl>CD36* (*n* = 70) embryos. Error bars denote standard error of the mean (SEM), p > 0.05 not significant (ns), **p < 0.005, ***p < 0.0005, and ***p < 0.0001 (unpaired two-tailed *t*-tests). Scale bars, (**A**) 50 μm, (**C, D**) 10 μm, and (**F**) 5 μm.

The online version of this article includes the following source data and figure supplement(s) for figure 1:

**Figure supplement 1.** Conserved function of Emp and its human homolog CD36.

**Figure supplement 1—source data 1.** This zip archive contains the raw unedited western blot shown in *Figure 1—figure supplement 1*.

## Dynamic subcellular localization of Emp during tracheal development

To further elucidate Emp functions, we generated an antibody against its extracellular domain (*Figure 1—figure supplement 1B*) and determined its subcellular localization by co-staining for the previously characterized apical membrane proteins, Crb and Ptp10D, markers of adherens junctions (DE-cad, pY) and septate epithelial junctions (Disc Large/Dlg, Coracle/Cora). During tracheal branch elongation (stage 14–16), *wild-type* embryos showed an Emp enrichment in epithelial apical membranes and in subapical cytoplasmic puncta (*Figure 2A, B*). Similar to the tracheal cells, Emp showed apical localization, initially diffuse in dots in subapical regions and progressively more defined at the apical cortical region in the epidermis, hindgut and tracheal terminal branches of stage 15–16 *wild-type* embryos (*Figure 2—figure supplement 1A, B* and *Figure 2—figure supplement 2*). During late stage 16 to early stage 17, Emp localization became predominantly restricted in the junctional subapical region of tracheal, epidermal, and intestinal cells, where it colocalized with Crb, DEcad-GFP, and Phospho-Src (*Figure 2A, C* and *Figure 2—figure supplement 1A–C*). The Emp signal showed only weak colocalization with the SJ markers Coracle, Mtf, Dlg, and with the lateral cytoskeleton marked by α-Spectrin (*Figure 2—figure supplement 1A–C*). The massive uptake of luminal material correlates with the disassembly of apical actin bundles running along the transverse tube axis. The diaphanous-like formin, DAAM and the type III receptor tyrosine phosphatases, Ptp4E and Ptp10D, control the organization of F-actin bundles running along the perpendicular tube axis in the *Drosophila* airways (*Matusek et al., 2006*; *Tsarouhas et al., 2019*). Mutations in *Ptp10D4E* (*Ptp10D* and *Ptp4E*) or expression of a dominant negative form of DAAM (*btl>C-DAAM*) disrupt the transverse actin bundle arrays and prematurely initiate luminal protein clearance. Similarly, Latrunculin A (Lat-A) injection in embryos destroys the actin bundles and leads to luminal protein uptake. These experiments had suggested that the transverse F-actin bundles restrict the endocytic uptake of luminal cargoes (*Tsarouhas et al., 2019*). We analyzed wild-type embryos over-expressing the GFP-tagged actin-binding domain of moesin (*btl>moe-GFP*) in tracheal cells stained for Emp and GFP. This showed low colocalization ($r^2$ = 0.218, *n* = 5) suggesting distinct apical membrane domains of Emp and actin bundles (GFP) (*Figure 2—figure supplement 3A–D*). We infer that Emp is predominantly localized in F-actin bundle-free membrane regions to promote the endocytosis and recycling of selected luminal cargos along the longitudinal tube axis. Next, we tested whether the localization of Emp may be altered in *Ptp10D4E* mutants and in embryos overexpressing the dominant negative C-DAAM construct in the airways. Embryos of both genotypes showed premature translocation of Emp to the airway cell junctions compared to *wild-type* (*Figure 2D, E*). This suggests, that similar to luminal protein uptake, the relocation of Emp to the apical junctional region can be induced by the premature disassembly of the actin bundles. Additionally, we analyzed the localization and intensity of Emp and Crb protein stainings along the apical membrane in *wild-type* and *btl>C-DAAM* embryos. DAAM inactivation increased the intensity of apical Emp punctate accumulations compared to *wild-type* (*Figure 2F, G*), further arguing that the transverse actin bundles restrict Emp localization at the apical membrane.

To test if a subset of the apical Emp puncta may correspond to endocytic vesicles, we analyzed the localization of Emp relatively to several YFP-tagged Rab GTPases (YRab), expressed at endogenous levels. Co-staining for Emp and GFP (*Dunst et al., 2015*) showed an overlap with YRab5 (early

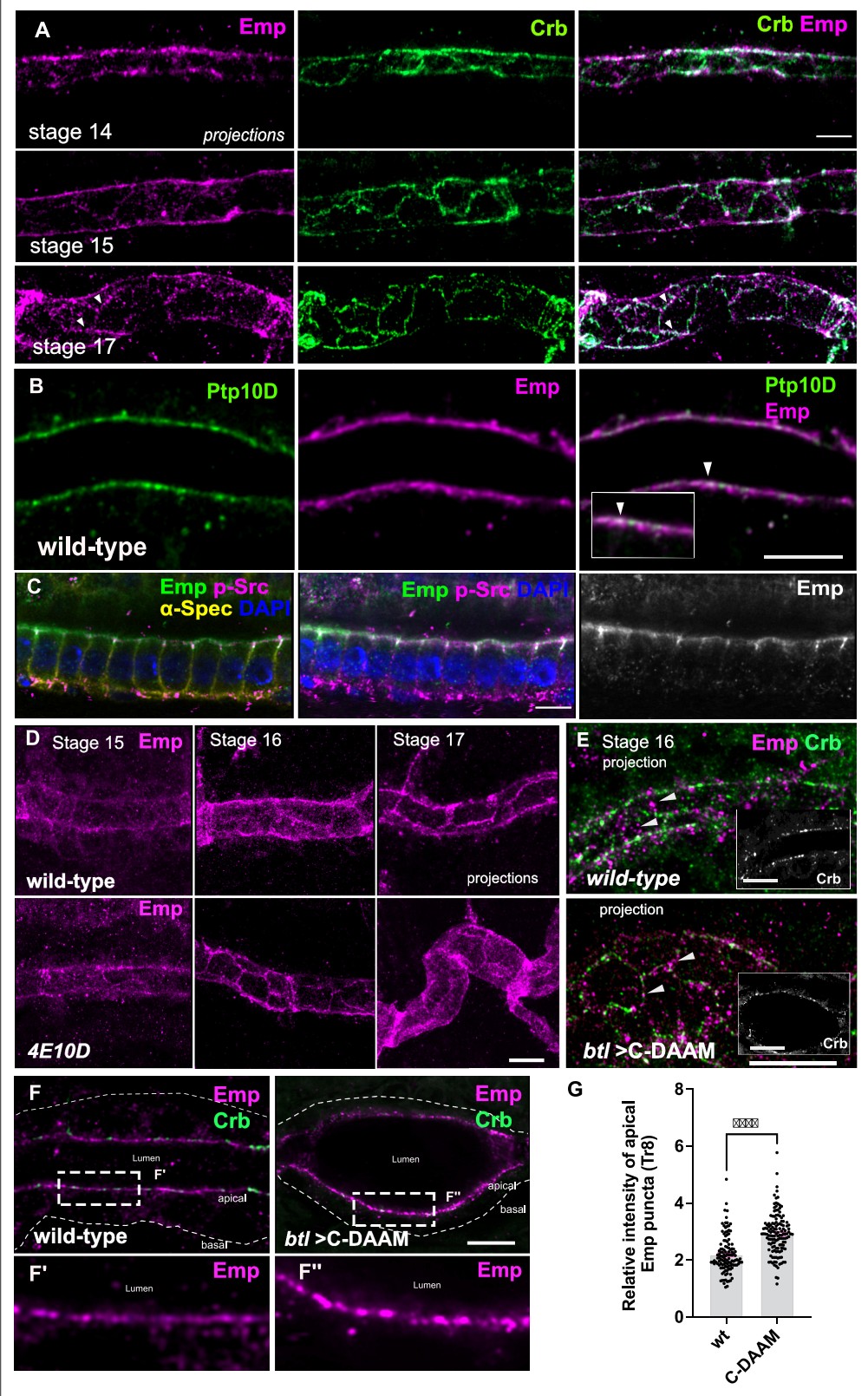

**Figure 2.** Dynamic apical distribution of Emp during tube maturation. (**A**) Confocal images showing dorsal trunk (DT) projections, from stage 14, 15, and 17 embryos, stained for Emp and Crb (Crumbs). (**B**) Confocal images of the DT in *wild-type* embryos stained for Emp and Ptp10D. Inset denotes a region of the apical membrane (arrowheads) with Emp and Ptp10D co-localization. (**C**) Confocal images of embryonic gut cells stained for Emp,

*Figure 2 continued on next page*

*Figure 2 continued*

p-Src, α-Spectrin (α-Spec), and 4',6-diamidino-2-phenylindole ( DAPI) showing the subcellular localization of Emp in stage 17. (**D**) DT projections, from stage 15, 16, and 17 of *wild-type* and *Ptp4E10D* (4E10D) stained for Emp. (**E**) Confocal image-projections of the DT cortex (depth: 5.2 μm) in *wild-type* and *btl>C-DAAM/+* (stage early-16) embryos stained for Emp and Crb. Emp is prematurely enriched at the AJs in *btl>C-DAAM/+*, but not in *wild-type* embryos (arrowheads). Images in insets are single longitudinal sections depicting the luminal borders of the tubes. Note: *btl>C-DAAM/+*, displays irregular tubes with ellipsoid dilations. (**F**) Confocal images showing the tracheal DT of *wild-type* and *btl>C-DAAM/+* embryos stained for Emp and Crb. Insets show magnified regions of (**F**), indicated by the white rectangles. The apical or basal sides of the tracheal cells are indicated. (**G**) Bar plot showing the relative intensity of apical Emp puncta at DT (Tr8) in *wild-type* (n = 168 puncta, 5 embryos) and *btl>C*-DAAM (n = 237 puncta, 6 embryos). Statistically significant shown in p-values, ***p < 0.0005 (unpaired two-tailed *t*-tests). Scale bars, 5 μm (**A, E, F**) and 10 μm (**B, C, D**).

The online version of this article includes the following figure supplement(s) for figure 2:

**Figure supplement 1.** Emp localization.

**Figure supplement 2.** The apical localization of Emp in dorsal trunk (DT) and terminal branches.

**Figure supplement 3.** Low apical co-localization of Emp with the cortical actin bundles.

endosomes) and YRab7 (late endosomes) with the Emp positive cytoplasmic puncta. We also detected weaker colocalization with YRab11 (recycling endosomes), (***Figure 3A***). Live imaging of embryos expressing *btl>Emp*-GFP in the time interval of luminal protein clearance (early stage 17) showed an increase of Emp intracellular puncta compared to stage 15 or late stage 17 embryos (***Figure 2— figure supplement 1D***). Overall, these experiments suggest that the localization of Emp in the apical membrane and endocytic vesicles is dynamic and influenced by actin bundle integrity. The timing of the final, steady-state accumulation of Emp is controlled by PTP signaling.

## Serp internalization and endosomal targeting requires Emp

The luminal retention of Serp in *emp* mutants and the partial localization of Emp with endosomal markers led us to examine if Emp mediates the endosomal uptake and subsequent trafficking of luminal Serp. We co-stained for endogenous YFP-tagged endosomal markers and Serp in *wild-type* and *emp* mutant embryos. This analysis showed that intracellular Serp puncta co-stained for the early endosomal marker Rab5 (***Figure 3A–C***; arrows, $r^2 = 0.29$) and late endosomal marker, Rab7 (***Figure 3A–C***; arrows, $r^2 = 0.27$) in *wild-type* embryos. In the *emp* mutants, the colocalization of Serp with both early and late endocytic markers was significantly decreased (***Figure 3B, C***), suggesting that Emp mediates Serp internalization and endosomal vesicle targeting.

In addition, the number of intracellular Serp puncta was reduced in *emp* mutant embryos compared to *wild-type* (***Figure 3—figure supplement 1A***), whereas the total number of GFP puncta corresponding to early and late endosomes remained unchanged (***Figure 3—figure supplement 1B***). The large size of YRab7 vesicles appeared reduced in *emp* mutants, presumably due to impaired apical internalization and targeting of cargoes to late endosomes. Because Emp also shares homology with the mammalian *lysosomal* integral membrane protein-2 (LIMP-2) (***Hart and Wilcox, 1993***), we cannot exclude an additional function of Emp in endo-lysosomal integrity and trafficking. Taken together, these results suggest that Emp functions as a receptor for Serp internalization and endosomal targeting.

To further investigate Emp cargo specificity, we tested the luminal clearance of GFP constructs, tagged with different domains of Serp in *wild-type* embryos and *emp* mutants (***Figure 3D***). We used GFP constructs fused to either Serp-full-length or to the Serp-LDLr-domain (low-density lipoprotein receptor-domain) or to the Serp-CBD-domain (chitin-binding domain) (***Luschnig et al., 2006***; ***Wang et al., 2006***) and examined their luminal secretion and clearance. The constructs were expressed and similarly secreted into the tracheal tubes of *wild-type* and *emp* mutant embryos. The Serp-LDLr reporter was cleared from the lumen efficiently as the full-length Serp-GFP but the CBD-GFP fusion was retained in the tracheal lumen of 20% of *wild-type* embryos. Interestingly, both the Serp-GFP and LDLr-GFP were retained in the lumen of *emp* mutants. These results suggest that the LDLr-domain of Serp, targets GFP to Emp-mediated internalization. CBD-GFP clearance failed in only 40% of the *emp* mutants suggesting that this cargo is also internalized by additional unknown receptors (***Figure 3E, F***). To further test if the addition of the LDLr-domain is sufficient to target an unrelated protein for Emp-mediated uptake, we fused the Serp LDLr-domain to the Gasp-mCherry protein, which does not

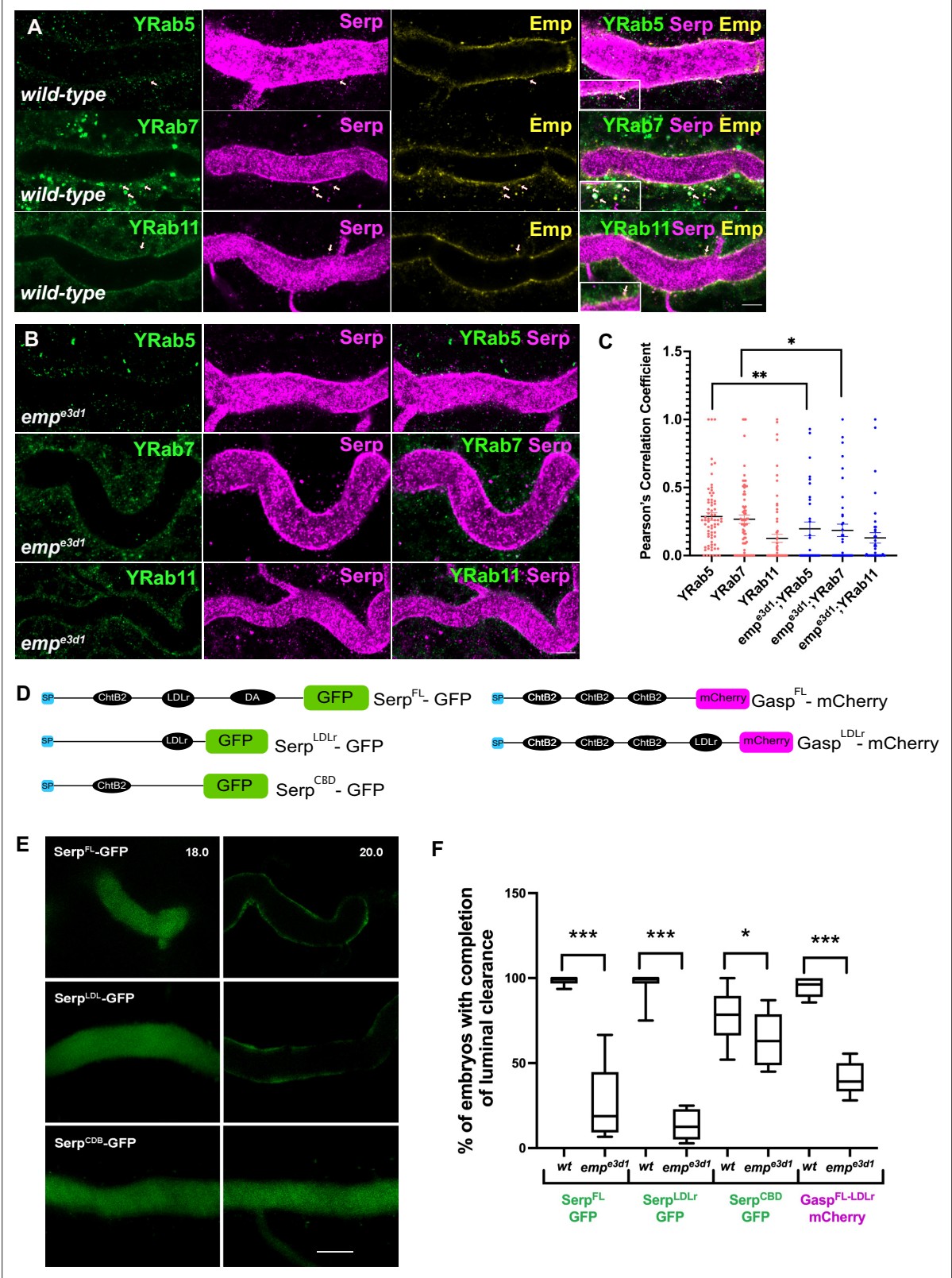

**Figure 3.** The endosomal localization of Serp is strongly reduced in *emp^e3d1* mutants. (**A**) Confocal images of tracheal dorsal trunk (DT) of *wild-type* embryos expressing endogenous tagged YFP-Rab (knock-in) proteins, YRab5, YRab7, and YRab11 stained with Emp, Serp, and GFP. Insets show zoomed cross section views of the DT (*y–z* plane) of Emp co-localization with YRab5, YRab7, and YRab11, as indicated with red arrowheads. (**B**) Confocal images showing the tracheal DT, stained for Serp and endogenous YRab5, YRab7, and YRab11 in *emp^e3d1* mutant. (**C**) Scatter plots representing

*Figure 3 continued on next page*

*Figure 3 continued*

the co-localization between the YRabs and Serp in *wild-type* and in *emp³ᵉᵈ¹* mutants calculated by Pearson correlation coefficient (*r²*). (**D**) Schematic representation of Serp and Gasp domain organization. The following abbreviations are used: SP, signal peptide (blue); LDLr, low-density lipoprotein receptor (black); ChtB, chitin-binding domain (black); GFP (green); Cht BD2, chitin-binding domain (black); and mCherry (magenta). *btl>SerpFL-GFP* represents the full length of Serp, *btl>SerpLDLr-GFP* represents the LDLr-domain of Serp, *btl>SerpCBD-GFP* expresses ChtB domain of Serp, *btl>GaspFL-mCherry* represents the full-length Gasp protein *and btl>GaspLDLr-mCherry* represents the full-length Gasp protein with addition of the LDLr-domain. (**E**) Confocal images showing the DT of live *btl>SerpFL-GFP, btl>SerpLDL-GFP, btl>SerpCBD-GFP* (green) embryos before (18.0 h AEL) and after (20.0 h AEL) luminal protein clearance. *btl>SerpCBD-GFP* embryos show incomplete luminal GFP clearance compared to *btl>SerpFL-GFP or btl>SerpLDLr-GFP*. (**F**) Plots showing the percentage of embryos with completion of luminal clearance in *btl>SerpFL-GFP* (green, n = 58); *empᵉ³ᵈ¹;btl>SerpFL-GFP* (green, n = 15); *btl>SerpLDLr-GFP* (green, n = 3 2); *empᵉ³ᵈ¹;btl>SerpLDLr-GFP* (green, n = 26); *btl>SerpCBD-GFP* (green, n = 45); *empᵉ³ᵈ¹; btl>SerpCBD-GFP* (green, n = 28); *btl>GaspFL-LDLr-mCherry* (magenta, n = 57); and *empᵉ³ᵈ¹;btl>GaspFL-LDLr-mCherry* (magenta, n = 56), from at least five independent experiments. The median (horizontal line) is shown in the plots with the data range from 25th to 75th percentile. Error bars denote standard error of the mean (SEM), *p < 0.05, ** p < 0.01, and ***p < 0.0005 (unpaired two-tailed *t*-tests). Scale bars, 5 μm (**A, B**) and 10 μm (**E**).

The online version of this article includes the following figure supplement(s) for figure 3:

**Figure supplement 1.** Quantification of Serp internalization.

require *emp* for its luminal clearance. As with the Serp-based constructs we analyzed the clearance of *Gasp^FL-mCherry* and *Gasp^FL+LDLr-mCherry* in *wild-type* and *emp* mutant embryos. Both constructs were normally cleared form the airways of *wild-type* embryos. However, *btl>GaspFL^FL+LDLr-mCherry*, but not *btl>Gasp^FL-mCherry*, was retained in the airways of *emp* embryos (**Figure 3F**). These data suggest that the LDLr-domain can confer cargo specificity for Emp-mediated internalization. Loss of function *verm serp* mutants or overexpression of *Serp-GFP* leads to over elongation of the tracheal tubes (**Luschnig et al., 2006**; **Wang et al., 2006**). We thus examined the effects of Serp and Verm on the levels and localization of Emp. *btl>Serp-GFP* overexpressing embryos showed increased punctate accumulations of Emp and Crb at the apical cell surface compared to *wild-type* (**Figure 4A**). Conversely, in *verm^ex245,serp^ex7* double mutants, we detected more diffuse punctate cytoplasmic accumulations for both Emp and Crb compared to *wild-type* (**Figure 4A, B**). To investigate whether the changes in intensity and localization of Emp and Crb puncta upon Serp-GFP overexpression or *verm* and *serp* deletion were due to changes in protein synthesis or stability we performed western blots of mutant and *wild-type* embryos. The total protein levels of Crb and Emp did not change in the mutants suggesting that the levels of luminal Serp control the punctate accumulation of Emp and Crb at the apical membrane (**Figure 4D–F**). As expected, overexpression of Emp in the airways (*btl>Emp*) increased the overall levels of cytoplasmic Emp, without a major influence on the accumulation of Crb (**Figure 4A–C**). Overall, these data suggest that the levels of luminal cargo control the punctate accumulation of Emp and Crb at the apical membrane. *btl >Serp-GFP* overexpression also causes DT tube over-elongation. This phenotype is similar to the loss-of-function phenotype of *verm serp* double mutants (**Wang et al., 2006**) and argues that deviations of the normal luminal Serp levels, too low or too high, can similarly cause tube over-elongation. Measurements from the Tr5 to Tr10 showed reduced DT elongation in *emp;btl>Serp-GFP* embryos compared to *btl>Serp-GFP*, suggesting that Serp-GFP overexpression control the length of the *Drosophila* airways, at least partially, through Emp (**Figure 4G**). Overall, these results suggest that Emp acts as a scavenger receptor for LDLr-domain proteins, such as Serp, thereby facilitating their internalization through clathrin-independent endocytosis. The levels of luminal Serp-GFP influence the localized apical membrane accumulation of Emp and Crb and interfere with tube elongation.

## Emp controls tube morphogenesis

Crb and DE-cad trafficking underlies the anisotropic growth of the apical surface and elongation of *Drosophila* airways (**Förster and Luschnig, 2012**; **Nelson et al., 2012**). We first quantified the ratio of longitudinal/transverse cell junction lengths (LCJ/TCJ) in *emp* mutants and *wild-type* embryos stained for Crb. The longitudinal junction length and aspect ratio (LCJ/TCJ) of *emp* mutant tracheal cells significantly increased, while the length of junctions along the transverse tube axis was not affected (**Figure 5A–C**).

The selective accumulation of the apical polarity protein Crb at longitudinal junctions relates to tube elongation (**Olivares-Castiñeira and Llimargas, 2018**). Consistent with this, we detected

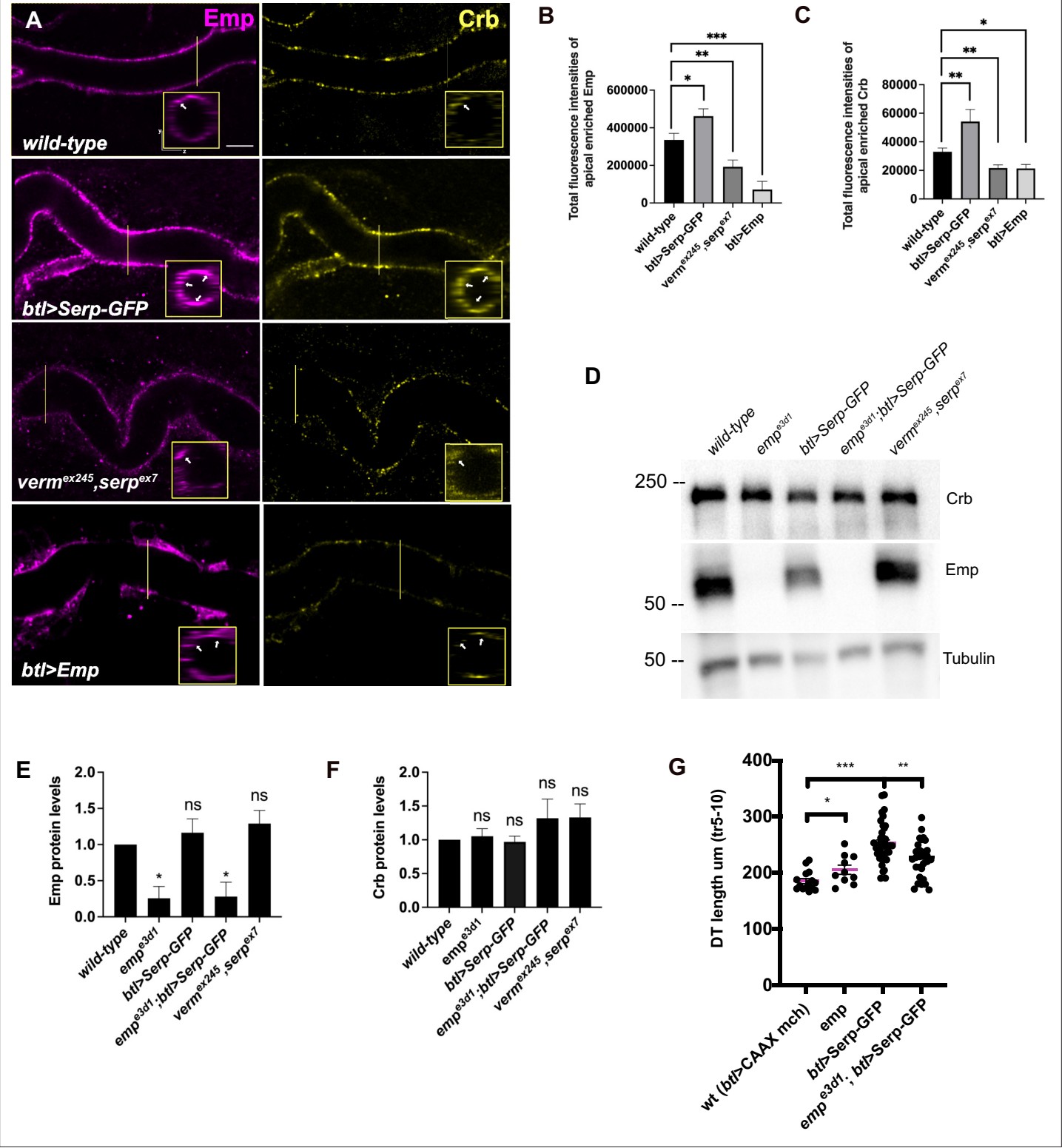

**Figure 4.** Serp-GFP overexpression induces apical Emp accumulations and tracheal over-elongation. (**A**) Confocal images stained for Emp and Crb, in *wild-type*, *btl>Serp-GFP*, *verm^ex245^,serp^ex7^* mutant, and *btl>Emp* embryos. Inset shows zoomed view of Emp and Crb signals. The arrows indicate the accumulation of Emp and Crb in *yz* plane. Bar plots showing total fluorescence intensities of apical enriched Emp (**B**) or Crb (**C**) in *wild-type* (*n* = 15), *btl>Serp-GFP* (*n* = 12), *verm^ex245^,serp^ex7^* (*n* = 10) mutant, and *btl>Emp* (*n* = 7) embryos. (**D**) Representative western blot from protein lysates of *wild-type*, *emp^e3d1^*, *btl>Serp-GFP*, *emp^e3d1^;btl>Serp-GFP* and *verm^ex245^,serp^ex7^* mutants, blotted for Emp and α-Tubulin. (**E**) and (**F**) show the quantification of protein

*Figure 4 continued on next page*

*Figure 4 continued*

levels of Emp and Crb, respectively, based on four independent western blot experiments (*n* = 4). (**G**) Plots representing the length of the tracheal dorsal trunk (DT) in μm (tr 5–10) from *btl>CAAX-mcherry* (control, *n* = 15), *emp* ^e3d1^ (*n* = 10), *btl>Serp-GFP* (*n* = 37) and *emp* ^e3d1^;*btl>Serp-GFP* (*n* = 29) embryos. Statistical significance shown in p-values; ****p < 0.0001, ***p < 0.0005, **p < 0.01, *p < 0.05, and p > 0.05 not significant (ns) (unpaired two-tailed *t*-tests). Scale bars, 5 μm.

The online version of this article includes the following source data and figure supplement(s) for figure 4:

**Source data 1.** This zip archive contains the raw unedited western blot shown in *Figure 4D*.

**Figure supplement 1.** Epithelial cell integrity is not affected in *emp*^e3d1^ mutants.

increased Crb signals along the longitudinal but not transverse junctions in over-elongated tubes of *emp* mutants (*Figure 5D–F*). Stainings for the AJ component DE-cad showed increased levels along both longitudinal and transverse junctions in *emp* mutants (*Figure 5D–F*). To examine whether the intensity differences may be due to changes in overall protein levels, we analyzed the relative protein levels of Crb and DE-cad in *emp* mutants by western blots. We observed no significant difference in total protein levels of Crb (*Figure 5GI*) or DE-cad (*Figure 5H, J*) in *emp* mutant embryos compared to *wild-type*. These results suggest that Crb accumulates in LCJ and DE-cad in LCJs and TCJs in *emp* mutants presumably due to defects in their endocytic uptake and trafficking. The relative LCJ vs. TCJ intensities (Raw Intensity/Junctional length) of stainings for the AJs component α-CAT (α-Catenin) (*Pai et al., 1996*) and the SJs protein Dlg (disc large) (*Olivares-Castiñeira and Llimargas, 2018*; *Sharifk-hodaei et al., 2019*) were indistinguishable between *emp* and *wild-type* embryos (*Figure 4—figure supplement 1A–D*), suggesting that intracellular junction components are intact in the mutants. Similarly, a dextran leakage assay comparing paracellular junction integrity of *wild-type*, *emp*, and *ATPα* mutant embryos showed that *emp* loss does not affect general SJ integrity (*Figure 4—figure supplement 1E*). Overall, these results suggest that Emp function regulates the apical membrane levels of Crb and DE-cad without majorly affecting junctional integrity or function.

## Emp modulates the apical actin organization

Our analysis until now suggests that Emp is a scavenger receptor involved in the continuous, basal-level trafficking of Serp during tube elongation and in its massive uptake during luminal protein clearance. Since the organization of the transverse actin bundles restrict both tube elongation and the timing of luminal protein clearance, we investigated the apical actin cytoskeleton in *emp* mutants. We first examined the apical F-actin in DTs of late stage embryos by live imaging using *btl>Moe-GFP*. At 17.5 h AEL, at the time interval of airway protein clearance, *emp* mutants showed a more diffuse and continuous apical actin bundle density compared to *wild-type* embryos (*Figure 6A, B* and *Figure 6—figure supplement 1E*). We also detected higher apical accumulation of the dDAAM formin, in *emp* mutants compared with *wild-type* embryos by antibody stainings. Stainings for Gasp that visualized the forming taenidial of ECM, did not show detectable defects in chitin deposition in the *emp* mutants (*Figure 6C*). These observations suggested that Emp modulates the actin bundle organization at the apical membrane, presumably upon engagement with its luminal cargoes.

To identify proteins that might provide a direct connection between the cytoskeleton and Emp, we used the cytoplasmic C-terminus of Emp (484–520 aa) as bait in a yeast two hybrid (Y2H) screen at Hybrigenics. This uncovered three potential interacting proteins, CG32506 encoding a RabGAP protein, CG34376 encoding a protein with a predicted Zinc Finger motif and βH-Spec/Kst (*Figure 6D*). We further characterized the interaction with βH-Spec/Kst because of its established role in the apical cytoskeleton organization in *Drosophila* epithelial tissues (*Thomas and Williams, 1999*; *Phillips and Thomas, 2006*). The α- and β-heavy Spectrin subunits form tetramers, which assemble in two dimensional networks together with actin underneath the apical plasma membranes. Additionally, βH-Spec/Kst is required for the early steps and endocytic trafficking of V-ATPase in the brush border of intestinal epithelial cells (*Phillips and Thomas, 2006*). We used a knock-in, fusion construct of Venus into the *kst* locus to detect endogenous, βH-Spec/Kst together with Emp and α-Spectrin in epithelial cells of the trachea and hindgut. As expected Kst-Venus colocalized with Emp apically, whereas α-Spectrin was also detected along the lateral sides (*Figure 6E* and *Figure 6—figure supplement 1D*). To confirm the Emp binding to Kst and also map their interaction domains we used immunoprecipitation experiments of tagged proteins in *Drosophila* S2 cells. We expressed V5-tagged Emp

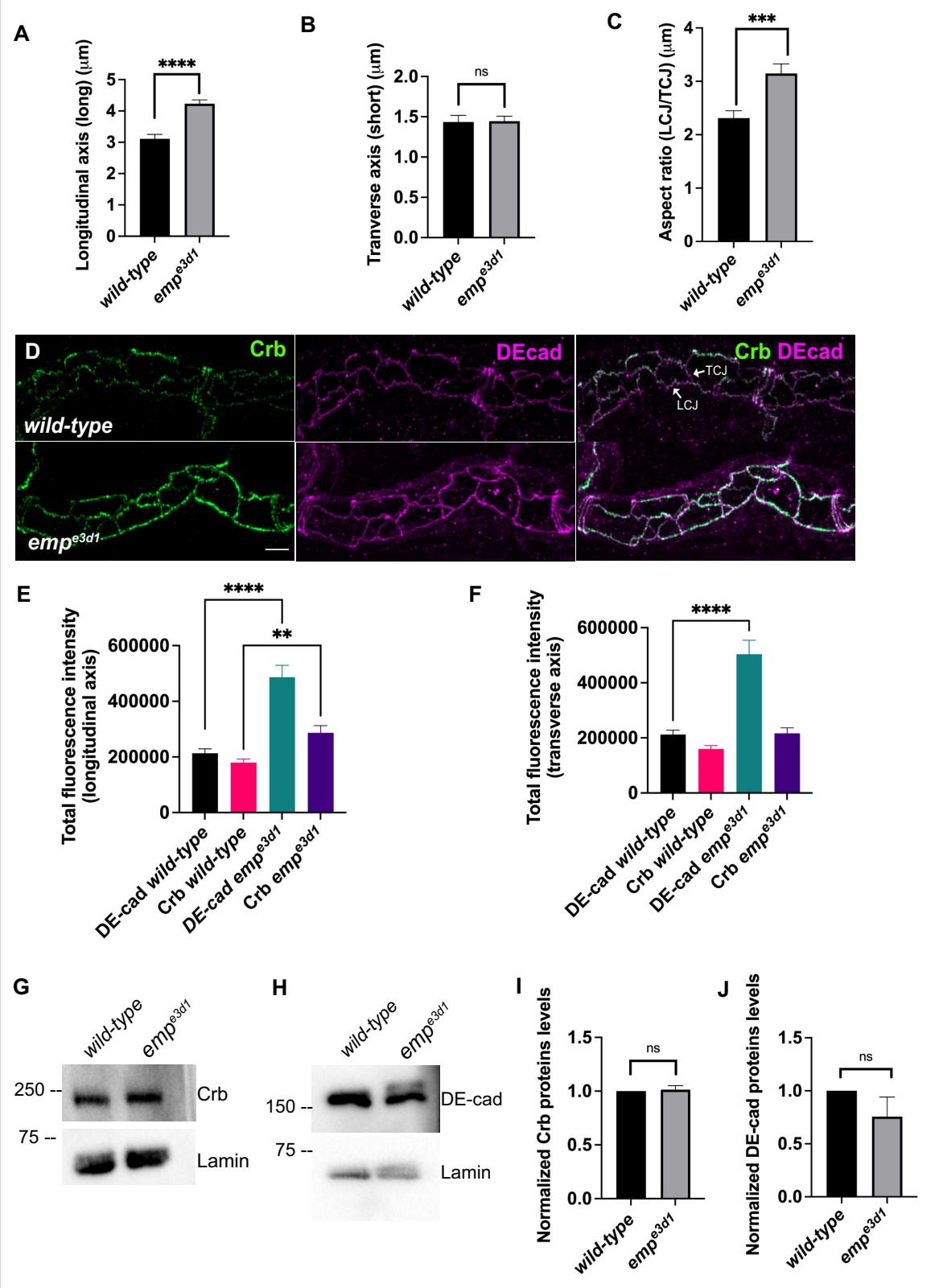

**Figure 5.** Emp modulates the Crb and DE-cad levels to control tracheal tube elongation. (**A**) and (**B**) bar plots showing the length (in µm) of longitudinal and transverse cell axis, respectively, in *wild-type* and *emp^e3d1* embryos. (**C**) Bar plots showing the aspect ratio, between longitudinal over transverse axis (LCJ/TCJ) in *wild-type* (*n* = 30), and *emp^e3d1* (*n* = 30), mutant embryos. ****p < 0.0001, ***p < 0.0005, and p > 0.05 not significant (ns) (unpaired two-tailed *t*-tests). (**D**) Projection images of *wild-type* and *emp^e3d1* mutant embryos, from stage 16, stained for Crb and DE-cadherin. (**E**) and (**F**) show

*Figure 5 continued on next page*

*Figure 5 continued*

quantifications of the fluorescence intensities of Crb and DE-Cadherin along longitudinal (**E**) and transverse (**F**) axis in *wild-type* (*n* = 78), and *emp*$^{e3d1}$ (*n* = 66), mutants. ****p < 0.0001 and **p < 0.01 (Mann–Whitney test). Representative western blot from protein lysates of *wild-type* and *emp*$^{e3d1}$ mutants, blotted with anti-Crb (**G**) or anti-DE-cad (**H**) and anti-Lamin (control). (**I**) and (**J**) are quantifications of Crb and DE-cad protein levels, respectively, based on three independent western blot experiments (*n* = 3). Statistical significance shown in p-values; p > 0.05 not significant (ns) (unpaired two-tailed *t*-tests). Scale bars, 5 μm.

The online version of this article includes the following source data for figure 5:

**Source data 1.** This zip archive contains the raw unedited western blot shown in *Figure 5G, H*.

and a series of constructs expressing different fragments of βH-Spec/Kst protein fused to the FLAG epitope. We found that the intracellular C-terminus of Emp co-precipitates with the C-terminal region of βH-Spec/Kst (*Figure 6F*) consistent with their interaction detected in the Y2H system. Additionally, *kst*$^2$ mutants show similar tracheal over elongation phenotypes with *emp* suggesting a functional interaction between Emp and Kst (*Figure 6—figure supplement 1A, B*). To further test this, we analyzed the localization and abundance of tagged βH-Spec/Kst in the airways of *emp* mutants. The apical levels of βH-Spec/Kst were severely reduced, while Crb staining was increased and staining of an unrelated apical protein Uninflatable remained unaffected in the *emp* mutants compared to the *wild-type* (*Figure 6G–I*). The localization or levels of Emp were not noticeably affected in *kst* mutants (*Figure 6—figure supplement 1C*). Together, these results indicate that the C-terminal intracellular domain of Emp binds to apical βH-Spectrin (Kst) and controls the spectrin cytoskeleton presumably by stabilizing βH-Spec/Kst. Additionally, in *emp* mutants the intensity and distribution of actin bundles is distorted and the levels of the diaphanous-related formin, DAAM are increased.

## Emp regulates Src phosphorylation

Src phosphorylation and activation are required for tracheal tube elongation and controls Crb accumulation at the longitudinal cell junctions (*Olivares-Castiñeira and Llimargas, 2018*), and the Beitel laboratory proposed that Src together with DAAM orient the directed membrane expansion during tube elongation (*Nelson et al., 2012*). Since *emp* mutants showed higher Crb accumulation in longitudinal junctions and an overall increase in the apical levels of DAAM, we compared Src phosphorylation (p-Src) levels by immunostainings and by western blot in *wild-type*, *emp* mutants, *btl>Serp-GFP* and *emp;btl>Serp-GFP* embryos (*Figure 7A–C*). The specificity of the p-Src$^{419}$ antibody was first confirmed by staining of *src42A src64B* double mutants (*Figure 7—figure supplement 1A*). Both immunostaining and western blot analysis showed increased p-Src levels in *emp* mutants compared to *wild-type* (*Figure 7A–C*). Interestingly, p-Src levels were also increased in the *verm serp* double or *verm serp* heterozygous (*Figure 7—figure supplement 1C, D*). Mutant embryos and decreased in *wild-type* embryos overexpressing *btl>Serp-GFP* (*Figure 7B*) This decrease by Serp overexpression was partly ameliorated in *emp* mutants (*Figure 7A–C*), suggesting that the effect of Serp-GFP overexpression on Src phosphorylation is, at least partly, mediated by Emp on the cell surface. We also detected increased p-Src levels in total protein extract from *emp* and *verm serp* mutant embryos (*Figure 7B*) suggesting that their interaction controls Src phosphorylation in other ectodermal tissues. In agreement with this, neurons of the ventral neve cord in *emp* mutant embryos also showed increased p-Src levels compared to *wild-type* (*Figure 7—figure supplement 1B*). To test if the increase in p-Src levels in *emp* mutants underlies the apical membrane over-elongation defects, we performed genetic interaction experiments between *src42A* and *emp* mutants. As expected, *emp* mutants showed over-elongated tubes while the *src42A*$^{E1}$ mutants showed short tubes (*Nelson et al., 2012*). The increase of tube length in *emp* mutants was significantly suppressed in *emp; src42A*$^{E1}$ embryos, where the levels of Src protein were reduced (*Figure 7D, E*). Similarly, the increased apical accumulation of Crb and DE-cad in *emp* mutants was partly restored by overexpression of a *Src42A*$^{DN}$ dominant negative form (*Src42A*$^{DN}$) in *emp* mutant embryos (*Figure 7—figure supplement 1E–H*). Previous work has shown that changes in chitin synthesis or ECM integrity affects the levels of phosphorylated Src42A (pSrc) at cell junctions (*Öztürk-Çolak et al., 2016*). Verm and Serp loss also causes p-Src upregulation and over-elongation and we also show that Serp overexpression leads to p-Src downregulation as expected (*Figure 7B*). Counter intuitively, Serp overexpression also results in tube over-elongation (*Wang et al., 2006*). Although Src42A is necessary and sufficient for cell shape changes during tracheal tube elongation (*Förster and Luschnig, 2012*; *Nelson et al., 2012*), this overexpression paradox could be

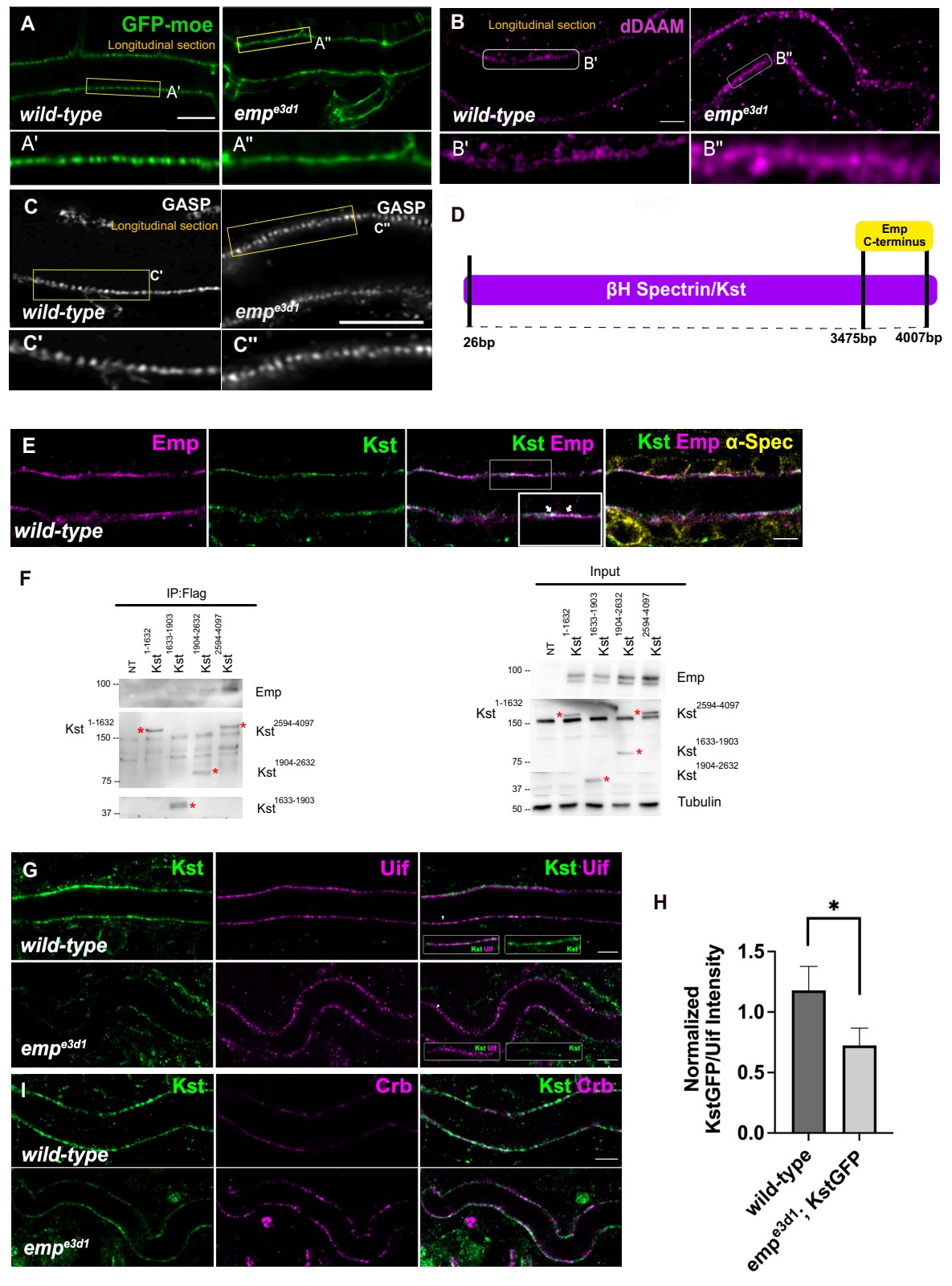

**Figure 6.** Loss of Emp affects apical F-actin organization. (**A**) Confocal images showing live dorsal trunk (DT) of *wild-type and emp^e3d1* mutant embryos from 17.5 h AEL expressing the actin reporter *btl>moe-GFP*. (**A', A"**) zoomed views of areas indicated by the rectangular frames of (**A**) panel. (**B**) Confocal images of dDAAM stainings in *wild-type* and *emp^e3d1* mutant embryos. (**B', B"**) shows magnified regions of (**B**), indicated with white rectangle. (**C**) Confocal images of GASP stainings showing the DT in *wild-type* and *emp^e3d1* mutant embryos at stage 17. (**C', C"**) are magnified regions of the

*Figure 6 continued on next page*

*Figure 6 continued*

apical extracellular matrix (ECM) as indicated by the rectangular frames in (**C**) panel. (**D**) Schematic view of the interaction domain of Emp C-terminus in 3475–4007 bp region of βH-Spectrin/Kst, as obtained by the Y2H screen. The tested Y2H prey-clones were covered the 26–4007 bps of the gene. (**E**) Confocal images showing the DT of Kst-Venus expressing embryos (*wild-type*) stained for Emp, GFP (kst), and α-Spec (α-Spectrin). (**F**) Co-immunoprecipitation of Flag-tagged Kst constructs from transfected S2 cells lysates, blotted with anti-Emp and anti-Flag. Input 2% is indicated. Red stars (*) denote the coresponding Kst band. st band. (**G**) Confocal images showing endogenous Kst-Venus in the DT of *wild-type* and *emp^e3d1^* mutants, stained for GFP (kst) and Uif (Uninflatable). (**H**) Bar plot showing Kst-GFP levels in *wild-type* (n = 5), and *emp^e3d1^* (n = 5), mutant embryos in relation to Uif. (**I**) Confocal images showing *wild-type* and *emp^e3d1^* mutant embryos stained for GFP (kst) and Crb. Statistical significance shown in p-value, * p < 0.05 (unpaired two-tailed *t*-tests) (**H**). Scale bars, 5 µm (**B, D, F, H**) and 10 µm (**A**).

The online version of this article includes the following source data and figure supplement(s) for figure 6:

**Source data 1.** This zip archive contains the raw unedited western blots shown in *Figure 6E*.

**Figure supplement 1.** Tube elongation is defected in *kst^2^* mutants.

explained by an indirect and src-independent effect of the overexpressed luminal Serp-GFP on tube elongation.

## Discussion

Emp, a CD36 homolog, is a selective scavenger receptor required for endocytosis of a subset of luminal proteins. The endocytosis defects in *emp* mutants become most apparent during the massive endocytosis wave that removes all secreted luminal components just before gas-filling of the airways. Our comparison of the endocytosis requirements of Serp and Gasp together with the LDLr-domain swap experiments defined a subset of Emp cargoes and indicates that the LDLr-domain targets cargoes to Emp through clathrin-independent endocytosis. These selective requirements of Serp and Gasp internalization are consistent with the view that the choice of endocytic route is a cargo-driven process (*Mettlen et al., 2018*).

The machinery involved in class B scavenger receptor endocytosis have not been characterized in *Drosophila*, but an important characteristic of all clathrin-independent endocytosis pathways is their dependance on the dynamic control of actin polymerization to distort the plasma membrane for cargo internalization (*Mayor et al., 2014*). Pioneering biochemical experiments suggested that CD36 receptor clustering is essential for its internalization and signaling in response to multivalent cargo binding. Single molecule tracking of CD36 in primary human macrophages showed that the un-ligated receptor diffuses in linear confinement tracks set by the actin cytoskeleton. These diffusion tracks enable clustering and internalization upon oxLDL-ligand addition (*Jaqaman et al., 2011*). Similarly, in human endothelial cells the actin cytoskeleton is required for the increase of CD36 clustering upon thrombospondin binding and Fyn, a Src-family kinase, activation (*Githaka et al., 2016*). These studies argue that CD36 is confined in cytoskeletal tracks and cargo/ligand binding induces its clustering and signaling ability. Similarly, to CD36, Emp function is also tightly connected with the apical cytoskeleton. First, *emp* activity is confined in apical 'macro'-domains along the longitudinal tube axis by the transverse, DAAM-dependent actin filaments. Disruption of these filaments induces massive endocytosis and re-localization of Emp. Additionally, the initially punctate Emp distribution along the apical membrane becomes aggregated upon overexpression of luminal Serp, suggesting that LDLr-domains on chitin-binding proteins induce Emp clustering. Apart from the Emp similarities to CD36, our results also reveal an unexpected direct function of Emp in organizing the apical Spectrin and actin cytoskeleton. The distribution of the DAAM-formin, the transverse actin bundle density and the apical accumulation of Kst are all disrupted in *emp* mutants. Because we showed that the conserved C-terminal intracellular domain of Emp binds directly to Kst, we propose that a central function of Emp, and possibly CD36, is to also directly organize the epithelial cytoskeleton. This notion is supported by the ability of human CD36 to partially rescue the *emp* mutant phenotypes when overexpressed in the *Drosophila* airways.

How can a scavenger receptor restrict epithelial tube elongation? In the interval of embryonic stages 13–16, regulated Src and DAAM activities promote tube elongation and the formation and alignment of the transverse actin bundles (*Matusek et al., 2006*). These bundles inhibit Emp internalization and restrict Serp endocytosis and trafficking to the longitudinal tube axis. We showed that Emp is directly linked to the apical βH-Spectrin cytoskeleton and is required for its assembly.

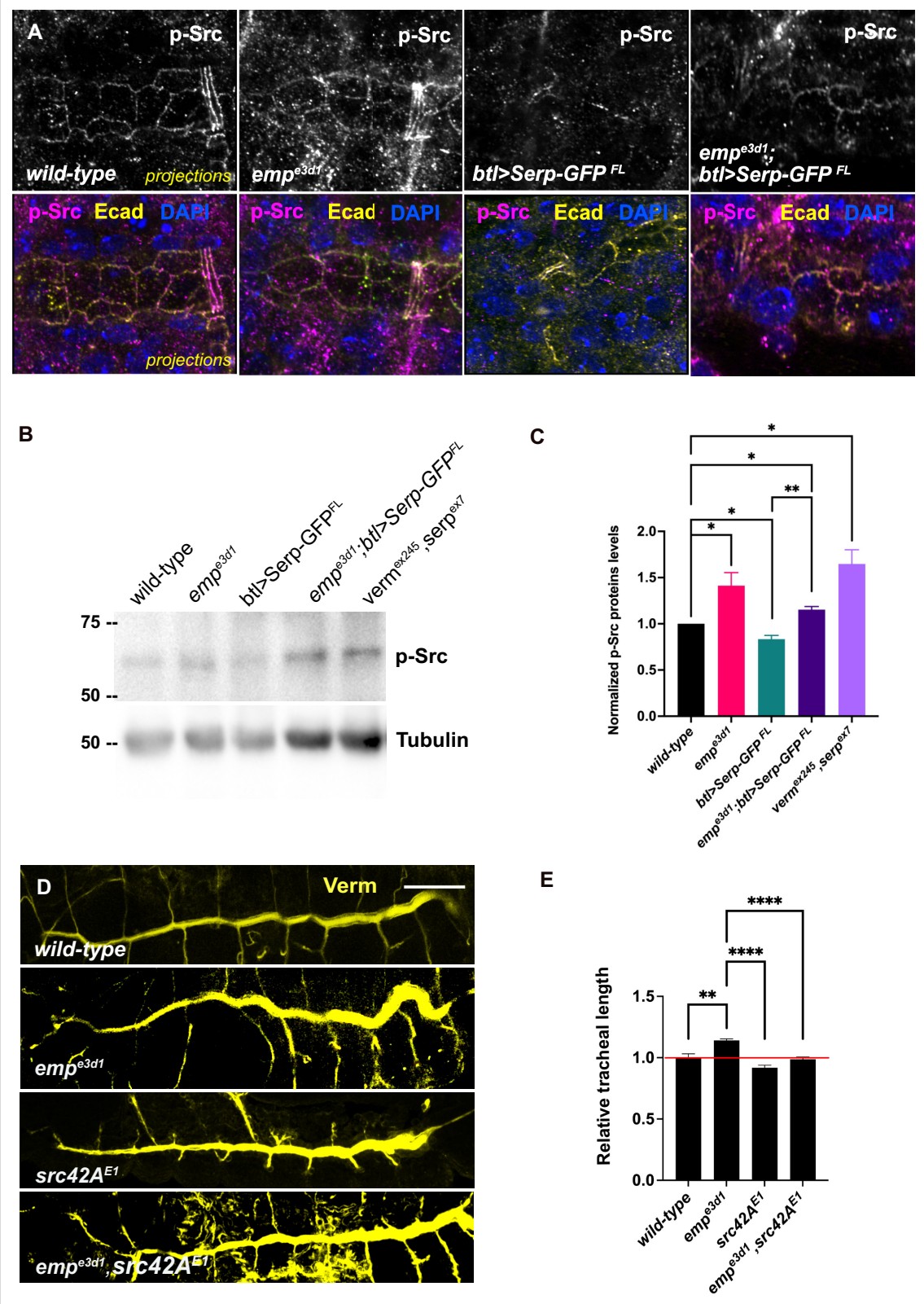

**Figure 7.** Elevated p-Src levels in *emp*[e3d1] mutants. (**A**) Confocal images showing projection of the tracheal dorsal trunk (DT) of *wild-type*, *emp*[e3d1] mutants, *btl>Serp-GFP* and *emp*[e3d1];*btl>Serp-GFP embryos* at late stage 16 to early stage 17, stained for endogenous p-Src. (**B**) Representative western blot from protein lysates of *wild-type*, *emp*[e3d1] mutants, *btl>Serp-GFP*, *emp*[e3d1];*btl>Serp-GFP* and *verm,serp* double mutant embryos, blotted with anti-p-Src and anti-α-Tubulin. (**C**) Quantifications of p-Src protein levels based on three independent western blot experiments ($n = 3$). *$p < 0.05$ and **$p <$

*Figure 7 continued on next page*

*Figure 7 continued*

0.01 (unpaired two-tailed *t*-tests). (**D**) Representative images of tracheal length size (DT), in *wild-type*, *emp^e3d1^*, *src42A^E1^*, and *emp^e3d1^,src42A^E1^* embryos stained for the luminal marker Verm. (**E**) Plots show the quantification of tracheal DT length of *wild-type* (n = 6), *emp^e3d1^* (n = 7), *src42A^E1^* (n = 10), and *emp^e3d1^;src42A^E1^* (n = 7) embryos. Error bars denote standard error of the mean (SEM) wild-type, *p < 0.05, **p < 0.005, and ****p < 0.0001 (unpaired two-tailed *t*-tests). Scale bars, 10 and 50 µm for images (**A**) and (**D**), respectively.

The online version of this article includes the following source data and figure supplement(s) for figure 7:

**Source data 1.** This zip archive contains the raw unprocessed western blots shown in *Figure 7B*.

**Figure supplement 1.** p-Src levels are increased in CNS of *emp^e3d1^* mutants.

This cytoskeletal organization might indirectly interfere with DAAM localization and presumably with Src42A activation and function as also proposed by *Nelson et al., 2012*. These data suggest that endocytosis of luminal proteins during tube elongation is controlled by at least two parallel pathways, one enabling Emp-endocytosis along the longitudinal tube axis and one restricting it along the transverse axis. We hypothesize that in the bundle-free membrane domains, Emp associates with Kst and presumably establishes an endocytosis domain, where Serp-mediated Emp clustering leads to internalization (*Figure 8A, B*). A similar function for Kst enabling endocytosis and trafficking of apical H⁺ V-ATPase has been proposed in the brush border of the larval *Drosophila* intestine, where its anchoring to the membrane remains unknown. The luminal levels of chitin deacetylases, like Serp, and other chitin modifying proteins are transcriptionally regulated (*Yao et al., 2017*) and are predicted to control the biophysical properties of the apical ECM (*Cui et al., 2016*). We infer that luminal Serp bound to chitin generates a multivalent ligand and is continuously recognized and endocytosed by Emp selectively along the longitudinal tube axis. Together with the clustered receptors, apical membrane and 'passenger' transmembrane proteins are expected to follow in the Emp endocytic vesicles. The over-elongation defects in *emp* mutant airways can be explained by the failure to balance elongation induced by Src activation with endocytosis of membrane and transmembrane regulators on the longitudinal tube axis (*Figure 8A, B*). Src42A phosphorylation levels have been proposed as a feedback mechanism of the aECM to the underlying cells to modulate proper cytoskeletal organization in airways (*Öztürk-Çolak et al., 2016*). Our work is complementary to this mechanism and further argues that Emp senses and responds to the levels of specific ECM cargoes by initiating apical membrane protein endocytosis and recycling during tube elongation. After protein clearance, Emp and Crb accumulate at the subapical region of the longitudinal junctions, where they could enable cytoskeletal interactions and the modulation of apical membrane tension.

At the initiation of protein clearance, the transverse bundles are transiently resolved and luminal proteins together with apical transmembrane proteins and membrane are massively internalized and targeted for degradation or recycling to the junctional areas (*Figure 8C*). The mechanism of Crb recycling in airway cells involves retromer components (*Olivares-Castiñeira and Llimargas, 2017*) but the selective targeting mechanism to longitudinal junctions is still unknown. The re-localization of Emp to the apical epithelial junctions during airway maturation is accompanied by massive alteration in Src activity, which is not restricted to the tracheal system but is common to several ectodermal organs including neurons. This suggests that scavenger receptor endocytosis has a general, but poorly understood role, in embryonic morphogenesis of epithelial tissues. Our work on Emp provides an entry toward elucidating the roles of scavenger receptor class B in development and pathogenesis.

## Materials and methods
### *Drosophila* strains

The *emp^e3d1^* null mutant was generated by FLP-FRT site-directed recombination using two piggyBac elements (PBacWH#021071 and PBacRB#e0441541). Embryos trans-heterozygous for Df(2R)BSC08 and *emp* were lethal showing identical phenotypes to *emp* homozygous embryos (*Supplementary file 1*). w^1118^ was used as the *wild-type* strain. In all experiments CyO and TM3 balancer strains carrying dfd-GFP were used to identify the desired genotypes. Flies were raised at 25°C and 50% humidity, with a 12-hr light–dark cycle.

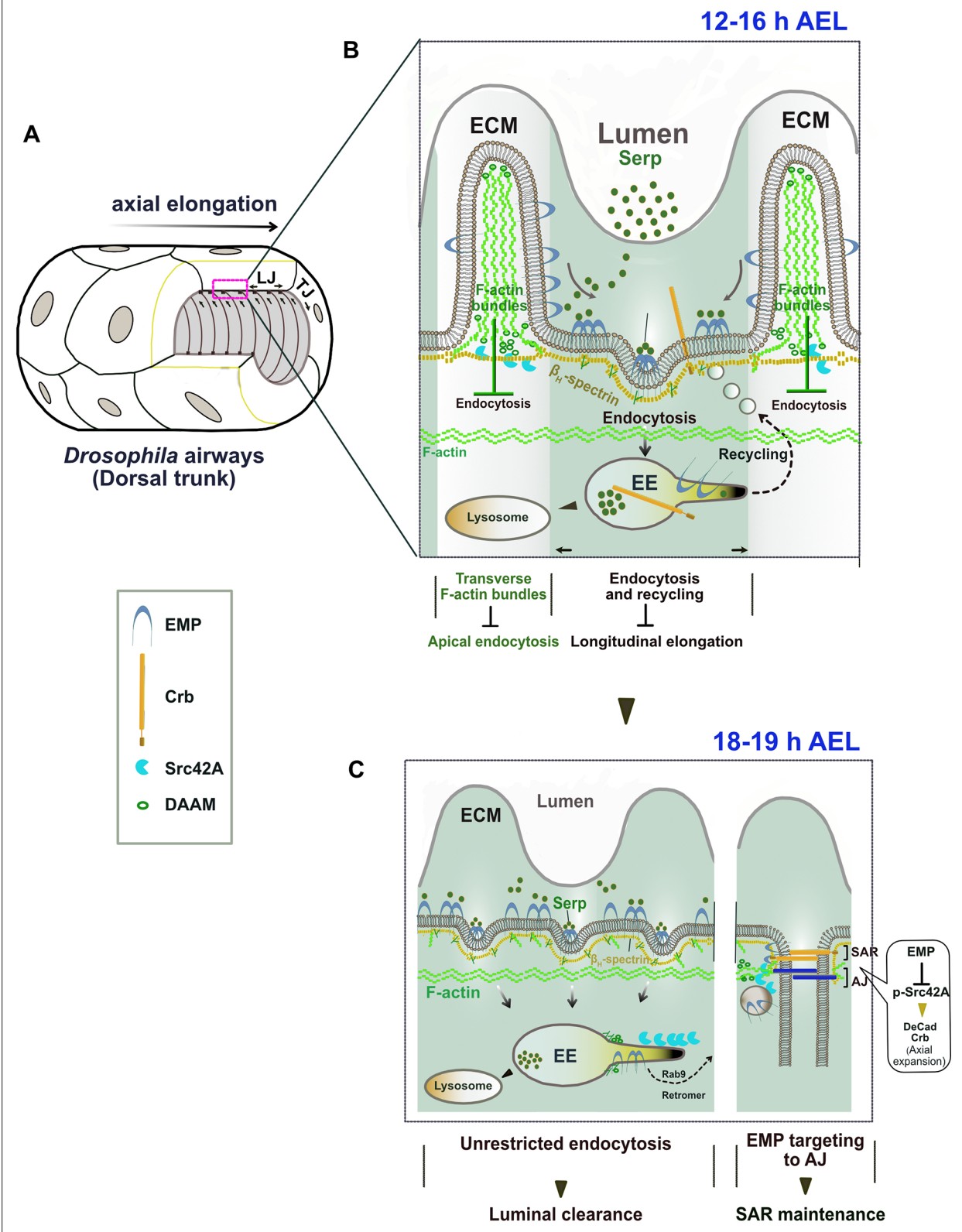

**Figure 8.** The proposed model of Emp regulation. (**A**) 3D drawing shows the multicellular tube (dorsal trunk) of *Drosophila* airways. The annular ridges of the apical extracellular matrix (ECM) are indicated in relation to the longitudinal elongation. LJ: Longitudinal junctions; TJ: transverse junctions. (**B**) Schematic zoom of the apical ECM region (**A**) showing the formation of two apical domains in the membrane–cytoskeleton interface of tracheal cells at 12–16 h AEL. Transverse F-actin bundles (white color zone) restrict apical Emp endocytosis along the transverse tube axis. In F-actin bundle-free

*Figure 8 continued on next page*

*Figure 8 continued*

membrane region (green color zone), Emp is associated with βH-Spectrin and Serp promotes Emp clustering leading to endocytosis and recycling of Serp and others 'passenger' transmembrane proteins (i.e. Crb) along the longitudinal tube axis. (**C**) Drawing of the apical, SAR and AJ regions during luminal protein clearance (18–19 h). Emp clears luminal Serp by endocytosis and trans-locates to the SAR/AJ to restrict p-Src42 activity and to control DE-Cad, Crb levels at the SAR/AJ.

## Molecular biology and transgenic flies

Complementary DNA (cDNA) encoding for *Human CD36* (RC221976) was cloned using Hifi DNA assembly kit (NEB E5520S) into the pJFRC-MUH vector (Addgene, plasmid 26213). bglII and XbaI were used to clone the amplified PCR fragment. *UAS-Emp* transgene was generated by cloning the cDNA sequencing of CG2727 by PCR amplification with primers containing EcoRI and KpnI. The *UAS-Gasp* construct was generated by sequential cloning of Gasp cDNA using primers containing EcoRI and XhoI. Further, LDLr-domain and mCherry were subcloned using enzymes XhoI and XbaI. The *UAS-Serp$^{FL}$-GFP* and *UAS-Serp$^{LDLr}$-GFP* constructs were generated according to *Luschnig et al., 2006*, and the *UAS-Serp$^{CBD}$-GFP* was provided by S. Luschning. *mCherry-Emp-V5His* construct was cloned using Hifi DNA assembly kit NEB E5520S into the pAc5.1/V5-His A vector (Invitrogen) using the following enzymes Acc65I and XhoI. The different Kst constructs were generously provided by N. Tapon. All the plasmids were confirmed by sequencing. Polyclonal antibody (anti-Emp) obtained by immunization with bacterially expressed recombinant polypeptides corresponding to amino acids 46–460 of Emp-A. Anti-sera obtained from immunized rats (Genscript).

## Co-immunoprecipitation and western blot analysis

*Drosophila* S2 cell extracts and Co-IP were prepared as previously described (*Tsarouhas et al., 2019*) and (*Fletcher et al., 2015*), respectively. The FLAG-tagged Kst constructs were provided by Nic Tapon (*Fletcher et al., 2015*). For detection of purified proteins and associated complexes, ChemiDoc XRS +system (Bio-Rad) was used. Western blots were probed with mouse anti-FLAG M2 (1:3000, Sigma, F3165), rat anti-Emp and rabbit anti-α-tubulin (1:2000, Cell Signaling, 11H10). For western blot analysis, *Drosophila* embryos were collected 12–20 h AEL and lysed in 20 µl of lysis buffer containing 50 mM HEPES (pH 7.6), 1 mM MgCl$_2$, 1 mM ethylene glycol-bis(β-aminoethyl ether)-*N,N,N',N'*-tetraacetic acid (EGTA), 50 mM KCl, 1% NP40, Protease inhibitor cocktail tablets (Roche #11697498001), and Phosphatase inhibitor cocktail 2 (Sigma-Aldrich #P5726). The lysates were centrifuged at maximum speed (30,060× *g*) for 10 min at 4 °C. Protein loading buffer (50 mM Tris/HCl, pH 6.8, 2% sodium dodecyl sulfate (SDS), 5% glycerol, 0.002% bromophenol blue) was added to the supernatant and samples were analyzed by SDS–polyacrylamide gel electrophoresis and immunoblotting according to standard protocols, using the ChemiDoc XRS +system (Bio-Rad), after application of the SuperSignal West Femto Maximum Sensitivity Substrate (Thermo Fisher Scientific, 34096). The following primary antibodies were used at the indicated dilutions: rabbit anti-α-tubulin (1:2000, Cell Signaling, 11H10), rabbit anti-Phospho-Src (1:750 Tyr418, Thermo Fisher), and Lamin ADL195 (1:100, DSHB).

## Quantification of western blots

For the western blot analysis, the actual signal intensity of each band of interest was estimated after subtraction of the background using ImageJ/Fiji software. The values were then divided by the corresponding intensity values of the loading control (α-tubulin or lamin).

## Immunostaining

Embryos were dechorionated in 5% bleach and fixed for 20 min in 4% formaldehyde saturated in heptane as described in *Patel, 1994*. The following antibodies were used: mouse anti-Ptp10D (1:10, 8B22F5, Developmental Studies Hybridoma Bank, DSHB), rabbit anti Phospho-Src (1:400, Tyr 419, Thermo Fisher), mouse anti Dlg (1:100, 4F3 DSHB), mouse anti-Crb (1:10, Cq4 DSHB), mouse anti-Coracle (1:100, C615.16 DSHB), DAAM antibody was a kind gift from József Mihály (*Matusek et al., 2006*), rabbit anti-GFP (1:400, A11122, Thermo Scientific), chicken anti-GFP (1:400, abl3970, Abcam), mouse GFP (1:200, JL-8, Clontech) gp anti-Verm (*Wang et al., 2006*), gp anti-Gasp (*Tiklová et al., 2013*), mouse anti-Flag M2 (1:3000, Sigma, F3165), Serp antibody were provided by S. Luschning (*Luschnig et al., 2006*). Secondary antibodies conjugated to Cy3 or Cy5 or Alexa Fluor-488 and -568

(Jackson Immunochemicals) were used and diluted as recommended by the manufacturer. For rat anti-DE-Cad (1:50, DSHB) embryos were fixed with 4% paraformaldehyde (PFA)–heptane for 20 min. Embryos expressing *moe-GFP* were dechorionated, devitellinized by hand and fixed in 4% PFA or formaldehyde (methanol free) in phosphate-buffered saline–heptane or PEM (PIPES-EGTA-MgCl2)–heptane mix, (PEM: 0.1 mM piperazine-N,N'-bis(2-ethanesulfonic acid) (PIPES) pH 6.9, 2.5 mM EGTA, 1 mM MgCl$_2$) for 20 min. Embryos were de-hydrated and mounted in pure methyl-salicylate (*Tran et al., 2016*). Stained embryos were imaged with an Airy-scan-equipped confocal microscope system (Zeiss LSM 800, Carl Zeiss) using a Plan-Apo ×63/1.40 DIC oil immersion objective.

## Yeast two-hybrid screen

The screen was carried out by HYBRIGENICS using a prey library constructed from RNA of embryos that were 0–24 hr old. A fragment encoding the C-terminus domain of Emp (amino acids 484–520) was inserted into the pB27 vector (N-LexA–bait-C fusion) and was used to screen 167 million clones.

## qPCR

Embryos dechorionated in bleach, hand-sorted for GFP expression, collected in 300 ml of TRizol LS Reagent and stored at −80°C until further use. For RNA extraction, the embryos were homogenized in TRIzol LS Reagent using a 1.5-ml tube pestle and the total RNA was purified using the Direct-zol RNA MicroPrep kit (R2060, Zymo Research). RNA was resuspended in RNase-free water and subsequently treated with DNAse I (AMPD1-1KT**,** Merk), for genomic DNA removal. Then 400 ng of RNA was reverse transcribed using High-Capacity RNA to cDNA kit (4387406 Thermo Fisher). The cDNA products were subsequently diluted 1:5 and 2 µl were used as a template in each qPCR reaction. qPCR was performed using iTaq Universal SYBR Green Supermix (Bio-Rad). Generation of specific PCR products was confirmed by melting-curve analysis. Ct values for all genes were normalized to the levels of *Rp49*. For data analysis, the delta-delta Ct values were applied. The sequences of primers used are provided in *Supplementary file 2*.

## Live imaging

Dechorionated embryos mounted in a glass-slide with a gas permeable membrane (*Tsarouhas et al., 2007*). Widefield live imaging performed to analyze protein clearance and gas-filling on embryos as described in *Tsarouhas et al., 2019*. For confocal live imaging, embryos were imaged with a scanning confocal microscope (LSM 780, or 800 Carl Zeiss) equipped with an Argon and an HeNe 633 laser using a C-Apochromat ×63/1.2 NA water objective. Z-stacks with a step size of 0.5–1.0 µm were taken every 6 min over a 3–8 hr period. For high-resolution confocal live imaging, an Airy-scan-equipped confocal microscope system (Zeiss LSM 800, Carl Zeiss) was used. Z-stacks (0.16–0.2 µm step size) were taken every 15 min over a 2–4 hr period using a Plan-Apo 63 ×/1.40 DIC oil or a C-Apochromat ×63/1.2 NA water objective (Zeiss). Raw data were processed with the Airy-scan processing tool available on the Zen Black software version 2.3 (Carl Zeiss). Images were converted to tiff format using the Zen Black or ImageJ/Fiji software.

## Morphometric analysis

Tube length measurements were conducted in embryos stained for the luminal markers Verm or Serp. Tracheal lengths were measured by tracing the length of the DT determined by the luminal markers using the freehand line selection tool of ImageJ/Fiji software. For metamere length measurements, we traced the DT length between the corresponding TC (transverse connective) branches. Longitudinal and transverse cell junctions were defined according to the angles from the DT axis (angle = 0°). Junctions with clear orientation angle 0° ± 30° or 90° ± 30° were defined as longitudinal or transverse, respectively. The rate of gas-filling calculated as the percentage of embryos with gas-filled tracheae divided by the total number of embryos analyzed.

## Dextran injections

For dye-permeability assays, 10 kDa Dextran-TR (Thermo Fisher Scientific) was injected into late stage 16 embryos (after the maturation of SJ) as described in *Jayaram et al., 2008*. Injections were performed in a microinjection system (FemtoJet, Eppendorf) coupled to an inverted fluorescence microscope (Cell Observer, Zeiss).

## Quantification of fluorescence intensity

The total fluorescence intensity signal of Emp, Crb, DE-cad, Dlg, and α-Cat was measured with ImageJ by manually drawing a 5-pixel line with the '*Freehand Line tool*', over the junctional region of the tracheal cells. The total fluorescence intensity in TCJ and LCJ of Crb and DE-cad was measured according to *Olivares-Castiñeira and Llimargas, 2018*. Background signals of 5-pixel line were subtracted from the intensities. For the quantification of fluorescent intensity on apical Emp positive puncta, 0.75 µm$^2$ squares on each puncture were defined as the regions of interests (ROIs). Mean fluorescent intensity within these ROIs was measured in Fiji. These values were divided by the corresponding fluorescence intensity observed in the ROI of the co-stained Crb puncta. Background signals of 0.75 µm$^2$ square-ROIs were subtracted from the intensities. Data collected from ImageJ were transferred to an Excel file for further analysis and plotting by GraphPad Prism.

## Statistical analysis

Statistical analysis was carried out using two-tailed *t*-test for unpaired variables unless indicated. The type of statistical test, *n* values and p-values are all provided in the figure legends. The experiments were replicated three to six times. All statistical analyses were performed using GraphPad Prism 9.1. The number of biological replicates for all the experiments is indicated in the figure legends. No explicit power analysis was used to estimate sample sizes for each experiment.

## Acknowledgements

We would like to thank Stefan Luschnig, Stefano De Renzis, Nic Tapon, Matthias Behr, the Bloomington *Drosophila* Stock Center, the *Drosophila* Genomics Resource Center (DGRC; IN), and the Developmental Studies Hybridoma Bank (DSHB; IA) for fly strains, clones, and antibodies. We thank the fly community that isolated, characterized or distributed mutant strains or antibodies. Special thanks to Flybase for the *Drosophila* genomic resources. We thank the Stockholm University Imaging Facility (IFSU). We thank the former ERASMUS student Claudia's Ctortecka, members of the M Mannervik, C Samakovlis (particularly Ryo Matsuda), Q Dai, S Åström, and Y Engström laboratories for comments and support during this project. This work was funded by the Swedish Research Council and the Swedish Cancer Society to CS; by the Magn. Bergvalls stiftelse and O E och Edla Johanssons vetenskapliga stiftelse to VT; and by the German Research Foundation (DFG), grant KFO309 (project number 284237345) to CS.

## Additional information

### Funding

| Funder | Grant reference number | Author |
|---|---|---|
| Vetenskapsrådet | | Christos Samakovlis |
| Cancerfonden | | Christos Samakovlis |
| O. E. och Edla Johanssons Vetenskapliga Stiftelse | | Vasilios Tsarouhas |
| Magnus Bergvalls Stiftelse | 2021-04453 | Vasilios Tsarouhas |
| Deutsche Forschungsgemeinschaft | KFO309 | Christos Samakovlis |

The funders had no role in study design, data collection, and interpretation, or the decision to submit the work for publication.

### Author contributions

Ana Sofia Pinheiro, Data curation, Formal analysis, Validation, Investigation, Visualization, Methodology, Writing – original draft; Vasilios Tsarouhas, Data curation, Formal analysis, Supervision, Validation, Investigation, Visualization, Methodology, Writing – original draft, Writing – review and editing; Kirsten André Senti, Conceptualization, Investigation; Badrul Arefin, Investigation; Christos Samakovlis,

Conceptualization, Data curation, Formal analysis, Supervision, Funding acquisition, Writing – original draft, Project administration, Writing – review and editing

### Author ORCIDs
Vasilios Tsarouhas http://orcid.org/0000-0002-2933-1351
Badrul Arefin https://orcid.org/0000-0003-1117-9125
Christos Samakovlis https://orcid.org/0000-0002-9153-6040

### Decision letter and Author response
Decision letter https://doi.org/10.7554/eLife.84974.sa1
Author response https://doi.org/10.7554/eLife.84974.sa2

---

## Additional files

### Supplementary files
• Supplementary file 1. The fly strains used in this study.

• Supplementary file 2. The sequences of primers used for qPCR analysis.

• MDAR checklist

### Data availability
All data generated or analyzed during this study are included in the manuscript and supporting files. Unprocessed western blots are provided as source data files in zip format.

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
