## [Editor Report]

In this important work, the authors convincingly show that the *Drosophila* scavenger receptor Emp (homologous to human CD36) senses and responds to the levels of its cargo apical ECM proteins and triggers the initiation of apical endocytosis, thereby regulating tube length via controlling Crumbs and Src. This work will be of broad interest to cell and development biologists as well as cancer biologists.

---

## [Decision Letter]

**Decision letter after peer review:**

Thank you for submitting your article "Scavenger receptor endocytosis controls apical membrane morphogenesis in the *Drosophila* airways" for consideration by *eLife*. Your article has been reviewed by 3 peer reviewers, and the evaluation has been overseen by a Reviewing Editor and Utpal Banerjee as the Senior Editor. The following individuals involved in the review of your submission have agreed to reveal their identity: Amin S. Ghabrial (Reviewer #1); Shigeo Hayashi (Reviewer #3).

Essential revisions:

1) One point of concern expressed by all three is the phenotype description and interpretation of Srp-GFP overexpression and its effect on Src. Since this is the critical point linking the endocytosis to the anisotropic tube elongation, the authors should be encouraged to address and clarify this.

2) The authors fail to clearly show the localization of Emp at the apical membrane and its connection to apical actin structures and chitinuous aECM.

3) Discussing how the junction-enriched Crumbs contribute to selective axial cell elongation will be desirable to expand the scope of this work.

4) Discussing a possible molecular mechanism linking Emp to pSrc distribution would contribute to a better insight into the regulation of tube length during embryonic tracheal development.

*Reviewer #1 (Recommendations for the authors):*

This manuscript makes a valuable contribution to our understanding of tubulogenesis and of the critical role of vesicle trafficking in this process. I have a number of questions that I would like the authors to address in a revision and several errors that they can correct to improve the clarity of the manuscript.

– p 5. does emp expression driven by 69BGAL4 rescue to adulthood?

– What does it mean that some embryos did clear their lumens but still (I assume) showed tracheal defects?

– Is it ultimately pSrc that is the target of Emp? is its distribution affected in all embryos examined (eg. a fully-penetrant defect)?

– A description of chc mutant phenotype would be helpful – is there overlap with the emp loss of function phenotype?

– What happens to the 25% of embryos with normal gas-filling? do they have tube length defects? How penetrant is the tube length defects?

– How does Emp localization look in unicellular tubes? and are unicellular tubes also unable to gas-fill?

– p 11 and 13.

However, btl>GaspFL+LDLr-mCherry, but not btl>GaspFL-mCherry, was retained in the airways of emp embryos (Figure 3F compared with Figure 1E).

BUT, GaspFL+LDLr is not shown in figure 3F. This needs to be added.

Downregulation of src activity and the levels of luminal Serp-GFP are sensed by Emp to control apical membrane protein endocytosis and trafficking, and thereby

elongation.

This is a bit confusing, can it be re-phrased in a wild-type context: whatever Emp is doing normally it is not via an overexpressed transgene (Serp-GFP)? Does pSrc fall just before the surge in endocytosis? Is that dependent upon Emp? does decreased luminal Serp put the breaks on endocytosis? What about verm, serp het embryos; 50% reduction in cargo, does this affect endocytosis? If the activity of Src is regulated downstream of Emp, what is the sensing that Emp is doing? Isn't Emp acting on Src (which seems to be what is shown in the model figure)?

*Reviewer #2 (Recommendations for the authors):*

In Figure 2C it is not clear what is shown in the panel of the btl>cDAAM. Luminal borders are not visible and, if the scale is the same, the tube seems overexpanded circumferentially (same in E'). Dorsal trunks are much more expanded than the control and previously shown for the same conditions (Figure 5 Matusek et al. 2006). Please revise these images.

The authors say "the number of intracellular Serp puncta were reduced in emp mutant embryos compared to wild-type (Figure 3—figure supplement 1A), whereas the total number of GFP puncta corresponding to early and late endosomes remained unchanged (page 11, lines 352-354). This is not clear at all if you compare Figures 3A and B on the green channel. In emp mutants large YRab7 vesicles are not present, even the ones not associated with Serp, suggesting that another late endosomal trafficking is impaired. This should be discussed by the authors.

In figure 3 the authors only show the clearance of SerpFL, SerpLDL, and SerpCBD in wt embryos. There are no panels showing the clearance of these constructs in emp mutants. The panels showing the clearance of GaspFL and GaspLDL in wt and emp embryos are not shown nor quantified. These experiments should be present in panels in Figure 3 figure supplement 1. A graph with quantifications of GaspLDL should be included.

I do not agree with the authors' conclusions on the experiment of overexpressing Serp in an emp mutant background (Figure 4). The authors say that: "reduced DT elongation in emp;btl>Serp-GFP embryos compared to btl>Serp-GFP, suggesting that Serp overexpression controls the length of the *Drosophila* airways at least partially through Emp." However, if Emp is a receptor for Serp in charge of its removal from the lumen, in conditions of Serp overexpression in the absence of Emp, we should see the increased length in emp;btl>Serp-GFP in comparison to btl>Serp-GFP, not decreased. I think the experiment should be: If Emp acts as a scavenger receptor for Serp, does overexpression of Emp rescue the tube elongation phenotype of Serp overexpression? The authors should explain why Emp downregulation improves the mutant phenotype induced by Serp overexpression.

The authors claim that Emp modulates the apical actin organization and that loss of Emp affects this F-actin organization. However, to analyse apical F-actin in DTs and compare control with emp mutants (Figure 6 A and B), the authors should present not just a median view of the tube lumen, but also show the cortex of the lumen in order to better analyze and visualize the F-actin bundles (as in Hannezo et al. 2015, Ozturk-Çolak 2016). In addition, this would give a better view of Emp localization in relation to the F-actin bundles. According to the model proposed Emp is clustered outside the F-actin-rich bundles. F-actin analysis in the cortex of the tubes together with Emp localization would allow for this analysis.

In the discussion, the authors state that "First, emp activity is confined in apical "macro"-domains along the longitudinal tube axis by the transverse, DAAM-dependent actin filaments." This is really not shown in the manuscript. So, either the authors clarify Emp localization in "apical macro domains" and its relationship with DAAM-dependent actin filaments, or this should be removed.

When discussing the model in figure 8, the authors say that "In the bundle-free membrane domains, Emp associates with Kst and establishes an endocytosis domain, where Serp mediated Emp clustering leads to internalization." This is not shown by their data and it should be made clear that this is a hypothesis and not a thesis.

Ozturk-Çolak et al. (*eLife* 2016) presented a model whereby there is feedback between the chitinous aECM and the underlying cells. They reported that chitin downregulation increases the levels of p-Src in tracheal cells. There is a tight connection between chitin and the F-actin bundles, so I believe it is essential that the authors analyse the chitinous structure in the DT of Emp mutants. In addition, it would be very interesting (but not essential) to analyse if the chitinous luminal filament is also internalised via Emp.

Furthermore, the feedback model proposed by Ozturk-Çolak et al. could be very well integrated into this new model. The link between the aECM and the underlying cells and the modulation of the feedback could be sensed by Emp. This should be discussed.

*Reviewer #3 (Recommendations for the authors):*

One potentially insightful result is the phenotype of Serp-GFP overexpression, which caused a reduction of pSrc (Figure 7A-C) and elevation of membrane Emp (Figure 4A). Although down-regulation of Src and upregulation of Emp in other contexts are associated with defective tube elongation, btl>Serp-GFP caused tube over elongation (Figure 4G). This apparent contradiction should deserve highlighted, and a deeper investigation may open a better understanding of the tube elongation mechanism. One potential clue could be to look at the anisotropy of junctional Crumbs in the btl>Serp-GFP embryos.

---

## [Author Response]

Essential revisions:1) One point of concern expressed by all three is the phenotype description and interpretation of Srp-GFP overexpression and its effect on Src. Since this is the critical point linking the endocytosis to the anisotropic tube elongation, the authors should be encouraged to address and clarify this.

Previous work has shown that changes in chitin synthesis or ECM integrity affects the levels of phosphorylated Src42A (pSrc) at cell junctions (Ozturk-Colak et al., 2016). Loss of Verm and Serp also causes p-Src upregulation and over-elongation and we also show that Serp overexpression leads to p-Src downregulation as expected (Figure 7b). Counter intuitively, Serp overexpression also results in tube over-elongation (Wang et al., 2006). This overexpression paradox could be explained by an indirect, src-independent effect of the overexpressed luminal Serp-GFP on tube elongation. Although Src42A is necessary and sufficient for cell shape changes during tracheal tube elongation (Forster and Luschnig, 2012; Nelson et al., 2012), there are additional src-independent pathways involving septate junction components affecting normal tube elongation (Laprise et al., 2010). The overexpression experiment is informative on the roles of Emp in shaping the apical membrane and cytoskeleton. But the tube overelongation phenotype, resembles the airway over-elongation of null *serp verm* mutants. We interpret that the luminal levels of Serp are tightly controlled and the tube elongation phenotype of UAS-Serp GFP reflects a dominant effect that could be due to interference with the luminal functions of Serp or it’s recycling back to the membrane or with the function of SJ components.

In the context of this work, we are not able to delve further on the dominant effect of Serp-GFP overexpression on tube elongation. All we can state with certainty is that the normal levels of Serp are required for optimal elongation and that the Serp-GFP overexpression causes a dominant effect, which is known (Wang et al., 2006). Overall, the novelty of our data and interpretations lies in that there are at least two parallel, spatially segregated, mechanisms converging on the apical cytoskeleton and on the regulation of apical cell shapes (please see Figure 8). One is defined by Src phosphorylation, which promotes the formation of F-actin bundles by DAAM (Matusek et al., 2006; Nelson et al., 2012). The other is defined by Emp clustering by Serp and its direct association with βHeavy-Spectrin to facilitate endocytosis and apical recycling (Figure 2 and Figure 2 suppl. 3, Figure 4A, Figure 6 and Figure 6 suppl-1). Our data propose that the two mechanisms are antagonistic, DAAM-bundles restrict the localization of Emp and endocytosis, and Emp clustering in response to normal cargo levels organizes the localization of the actin bundles. It is possible that excessive Emp clustering and/or the interference with the ECM integrity (due to the deacetylase activity of Serp) upon Serp-GFP overexpression may cause additional dominant effects on Src activation, ECM assembly, protein endocytosis, protein and membrane recycling and thereby interfere with tube elongation. We hypothesize that the effects of *verm serp* loss of function (heterozygous or homozygous) and UAS-Serp GFP overexpression on Src activation and tube elongation are significant, but partly mediated by Emp endocytosis. This is supported by the quantification of the p-Src signals in Figure 7 and Figure 7—figure supplement 1. We have added few sentences addressing this issue in the text (lines: 500-504 and 723-733).

2) The authors fail to clearly show the localization of Emp at the apical membrane and its connection to apical actin structures and chitinuous aECM.

We are not sure that we understood the meaning of “clearly show” without a concrete example of what experiment would constitute clear evidence for the localization of Emp in the apical membrane. Nevertheless, to address this point, we additionally co-stained wild-type embryos with anti-Emp and anti-Ptp10D (an apical transmembrane protein) and also wild-type embryos expressing the apical membrane reporter UAS-CD4-Tom with anti-Emp and anti-RFP. These new experiments showed strong co-localization of Emp with Ptp10D and CD4-Tom in the apical membrane (Figure 2B and Figure 2 suppl.2) by high resolution confocal microscopy. In the first version, we had shown that Emp colocalizes with the apical trans-membrane protein Crb (Figure 2A) and the Zonula Adherens (ZA) marker pY (Figure 2 Suppl. 1C). Emp also directly binds and colocalizes with apical cytoskeletal organizer βHeavy-Spectrin (Figure 6E). Our analysis, and given that Emp contains two trans-membrane domains, strongly supports the apical membrane localization of Emp in the airways. Typically, an endocytosis scavenger receptor is also internalized in the endosomal compartment and recycled to the membrane or degraded.

To relate Emp localization to the apical actin bundles, we have co-stained wild type embryos over-expressing a GFP-tagged version of the actin binding domain of moesin (*btl*>moe-GFP) with anti-Emp and anti-GFP in tracheal cells. We have shown earlier that moe-GFP faithfully labels the cortical actin bundles, running perpendicular to the tube axis, during tube maturation (Tsarouhas et al., 2019). Analysis of *xz* or *yz* sections within Z-stacks of airy-scan confocal images in these embryos showed low colocalization (*r^2^* = 218, *n* = 5) and predominantly complementary pattern of apical Emp and actin bundle domains (GFP). These results further support our interpretation that Emp, localizes in the bundle-free membrane regions in physical association with βHeavy-spectrin to promote the endocytosis and recycling of selected cargos. The new data are shown in Figure 2 suppl. 3A-D.

Regarding the last point of connections between Emp and its cargo Serp, we had shown with high resolution confocal microscopy that the receptor and its cargo, Serp, colocalize in common endosomes. This is disrupted in *emp* mutants, and additionally genetic manipulations of the cargo or the receptor changes the localization of each other (Figure 3A-C), suggesting that the connections between Emp and Serp are functionally significant.

3) Discussing how the junction-enriched Crumbs contribute to selective axial cell elongation will be desirable to expand the scope of this work.

In our manuscript, we view apical membrane, Crb or Ecad as passive cargoes of Serp/Emp endocytosis and recycling in the actin cable-free regions along the longitudinal tube axis (Figure 8 legends). As previously proposed Crb may change the apical actin cytoskeleton and its tensile properties at the junctions. This is very interesting but beyond our focus.

4) Discussing a possible molecular mechanism linking Emp to pSrc distribution would contribute to a better insight into the regulation of tube length during embryonic tracheal development.

We have not detected any direct molecular link between Emp and Src phosphorylation. In a simple model, p-Src levels during tube elongation would be controlled by the levels of RTK signaling including Stit, InR, EGFR, PTP (Tsarouhas et al., 2014; Wang et al., 2009) (Tsarouhas et al., 2019), (Olivares-Castineira and Llimargas, 2017). p-Src contributes to DAAM-mediated actin cytoskeletal remodeling, tube elongation, and the construction of the perpendicular actin rings (Matusek et al., 2006b; Nelson et al., 2012). We have shown before that these perpendicular structures inhibit endocytosis on the transverse axis (Tsarouhas et al., 2019). The levels of luminal Serp might be sensed by Emp in the apical membrane and Emp clustering induces apical membrane endocytosis. This apical membrane internalization constantly balances actin polymerization by DAAM activity on the bundles and tube elongation. Parts of the membrane and TM proteins are recycled to the apical junctions. We showed that Emp is directly linked to the βH-Spectrin cytoskeleton and is required for its assembly. This cytoskeletal organization indirectly interferes with DAAM localization and presumably Src activation/function as also proposed by (Nelson et al., 2012). We have tried to incorporate the interpretation of our results in the Figure model (Figure 8). We have added a few more sentences in the discussion to clarify this (lines: 799-802 and 820-827).

Reviewer #1 (Recommendations for the authors):This manuscript makes a valuable contribution to our understanding of tubulogenesis and of the critical role of vesicle trafficking in this process. I have a number of questions that I would like the authors to address in a revision and several errors that they can correct to improve the clarity of the manuscript.– p 5. does emp expression driven by 69BGAL4 rescue to adulthood?

We have not examined the survival to adulthood of this genotype. However, we have seen viable 2nd instar larvae of *emp;69BGal4/UAS-Emp*. In addition, we have examined the lethality of *emp;btlGal4/UAS-Emp* embryos. Hatching larvae of this genotype die as 3rd instar suggesting that Emp is essential in other tissues.

– What does it mean that some embryos did clear their lumens but still (I assume) showed tracheal defects?

We assume that the reviewer refers to 25% of embryos with normal protein clearance and gas filling. These embryos show longer tubes. This might suggest that there may be an additional scavenger receptor required for protein clearance. There are 13 paralogs encoded by the fly genome. One of them is debris buster, which has been implicated in airway clearance (Wingen et al., 2017). We have re-phrase our statement in the new version.

– Is it ultimately pSrc that is the target of Emp? is its distribution affected in all embryos examined (eg. a fully-penetrant defect)?

We did not detect any direct link between Emp and p-Src levels. Src activation is a functionally relevant aspect of the phenotypes but we don’t have any evidence that Emp biochemically controls Src phosphorylation.

Both immunostaining and western blot analysis showed increased p-Src levels in *emp* mutants compared to wild-type (Figure 7A-C). This is a fully-penetrant phenotype at stage 17. Additionally, p-Src levels were also increased in the verm serp double mutant embryos and decreased in embryos overexpressing *btl>Serp-GFP* (Figure 7B). This decrease by Serp overexpression was partly improved in *emp* mutants (Figure 7A-C), suggesting that the effect of Serp-GFP overexpression on Src phosphorylation is, at least partly, mediated by Emp.

– A description of chc mutant phenotype would be helpful – is there overlap with the emp loss of function phenotype?

Yes, there is an overlap of the chc and the emp phenotypes in the trachea. *chc^1^* null mutant embryos have defects both in luminal protein and liquid clearance and tracheal tube length (Behr et al., 2007) (Tsarouhas et al., 2007) similar to emp mutants. We have added this to the references and to the text.

– What happens to the 25% of embryos with normal gas-filling? do they have tube length defects? How penetrant is the tube length defects?

We addressed this above. Tube length phenotype is fully penetrant (Figure 1B, n = 17).

– How does Emp localization look in unicellular tubes? and are unicellular tubes also unable to gas-fill?

We thank the reviewer for the interest. We analyzed unicellular tubes of *btl*>CD4-Tomato embryos stained for Emp and RFP. We initially detected diffused Emp in cytoplasmic dots (stage 15) but then at stage 16 Emp was progressively defined in the apical region of terminal branches. This localization pattern of Emp looks similar to the one we detected in the DT. These new data are shown in Figure 2 suppl. 2 B.

The gas filling defects were also present in unicellular tubes.

– p 11 and 13.However, btl>GaspFL+LDLr-mCherry, but not btl>GaspFL-mCherry, was retained in the airways of emp embryos (Figure 3F compared with Figure 1E).BUT, GaspFL+LDLr is not shown in figure 3F. This needs to be added.

Thanks for pointing out this error. We have now included the correct graph in Figure 3F.

Downregulation of src activity and the levels of luminal Serp-GFP are sensed by Emp to control apical membrane protein endocytosis and trafficking, and therebyelongation.This is a bit confusing, can it be re-phrased in a wild-type context: whatever Emp is doing normally it is not via an overexpressed transgene (Serp-GFP)?

We apologize for the confusion. We have revised this sentence.

Does pSrc fall just before the surge in endocytosis? Is that dependent upon Emp? does decreased luminal Serp put the breaks on endocytosis?

We observed a clear p-Src decrease at junctions concurrently with an accumulation of Emp just after luminal protein clearance (at mid-late stage 17), However, it was impossible to determine the initiation point of the pSrc decrease by stainings. There is no available live imaging reporter to precisely quantify pSrc activity in living embryos. The p-Src decrease is dependent on Emp since *emp* mutants show strong increase in p-Src levels during this interval. Luminal Serp levels are a key aspect of Emp internalization. Our data are consistent with the model, where increased Serp accumulation during development is necessary to induce Emp receptor clustering, which culminates with luminal protein clearance. Decrease of Serp or Verm levels would lead to slower endocytosis.

What about verm, serp het embryos; 50% reduction in cargo, does this affect endocytosis? If the activity of Src is regulated downstream of Emp, what is the sensing that Emp is doing?

We analyzed verm, serp heterozygous embryos co-stained for Gasp and pSrc. These embryos showed about 50% reduced p-Src levels compared to the verm serp homozygous double mutants (New panels, Figure. 7 – Supplementary 1C-D). This together with the Serp loss of function mutant analysis support that Serp levels are important for Src activation.

Isn't Emp acting on Src (which seems to be what is shown in the model figure)?

We propose that Emp indirectly interferes with Src activation. Emp directly associates and organizes the apical β heavy-sprectrin cytoskeleton. This also increases apical DAAM formin levels and enhances its function in F-actin polymerization. Formins generate linear actin polymers, whereas the Arp 2/3 complex enhances branched actin polymerization required for apical endocytosis (Figure 6 and Tsarouhas et al., 2019). Earlier work shows that DAAM binds and downregulates Src (Nelson et al., 2012). Consistent with this, *emp* mutants showed higher DAAM and p-Src signals. We have addressed this in the discussion of the current version (lines: 799-802). See also the response of editor's comment 4.

Reviewer #2 (Recommendations for the authors):In Figure 2C it is not clear what is shown in the panel of the btl>cDAAM. Luminal borders are not visible and, if the scale is the same, the tube seems overexpanded circumferentially (same in E'). Dorsal trunks are much more expanded than the control and previously shown for the same conditions (Figure 5 Matusek et al. 2006). Please revise these images.

We apologize for the confusion. The Figure 2C (in the current version, 2E) showing maximum intensity Z-projections in the apical cortex with identical Z-depth of 5.2 μm in both genotypes. The luminal borders are not expected to be seen in the mutant. The *btl* >c-DAAM expression causes irregular tubes with strong dilations of ellipsoid shape and with irregular actin bundles making difficult to capture the tube shape. We have reported these phenotypes earlier (Figure S7 C-F in (Tsarouhas et al., 2019)). In the current paper, we showed the Z-projections instead of longitudinal sections to convey the localization of Emp at AJ on the apical surface (see Author response image 1). At this Z-thickness the luminal borders cannot be captured in *btl* >cDAAM due to overexpanded tubes (See also reviewer's #3 response). We provide now (as insets) single longitudinal sections (Figure 2E) showing the luminal borders of the tubes.

**Author response image 1. sa2fig1:** Schematic overview of the imaging in the apical surface of the tubes. Single sections (upper row) and Z-stack maximum projections (lower row).

The authors say "the number of intracellular Serp puncta were reduced in emp mutant embryos compared to wild-type (Figure 3—figure supplement 1A), whereas the total number of GFP puncta corresponding to early and late endosomes remained unchanged (page 11, lines 352-354). This is not clear at all if you compare Figures 3A and B on the green channel. In emp mutants large YRab7 vesicles are not present, even the ones not associated with Serp, suggesting that another late endosomal trafficking is impaired. This should be discussed by the authors.

We agree with the reviewer. Although the total number of YRab7 vesicles is not significantly affected in *emp* mutants the size of YRab7 vesicles appear reduced. We believe that this is an indirect effect of the apical endocytosis phenotype of the emp mutants as less endocytosed membrane is supplied to the endo-lysosomal system leading to smaller vesicles. We have considered reviewer's comment and we revised the text in the "Results section" (lines: 421- 427).

In figure 3 the authors only show the clearance of SerpFL, SerpLDL, and SerpCBD in wt embryos. There are no panels showing the clearance of these constructs in emp mutants. The panels showing the clearance of GaspFL and GaspLDL in wt and emp embryos are not shown nor quantified. These experiments should be present in panels in Figure 3 figure supplement 1. A graph with quantifications of GaspLDL should be included.

Thank you for pointing out this error. We have now included the correct graph in Figure 3F.

I do not agree with the authors' conclusions on the experiment of overexpressing Serp in an emp mutant background (Figure 4). The authors say that: "reduced DT elongation in emp;btl>Serp-GFP embryos compared to btl>Serp-GFP, suggesting that Serp overexpression controls the length of the *Drosophila* airways at least partially through Emp." However, if Emp is a receptor for Serp in charge of its removal from the lumen, in conditions of Serp overexpression in the absence of Emp, we should see the increased length in emp;btl>Serp-GFP in comparison to btl>Serp-GFP, not decreased. I think the experiment should be: If Emp acts as a scavenger receptor for Serp, does overexpression of Emp rescue the tube elongation phenotype of Serp overexpression? The authors should explain why Emp downregulation improves the mutant phenotype induced by Serp overexpression.

Thank you for the comment. We agree that the effects of overexpression of Serp-GFP in wild type embryos and in *emp* mutants are difficult to interpret. Even though, lack of Emp in Serp overexpression improved the tube size phenotype the tracheal network still remained over elongated. This suggests that overexpressed Serp-GFP interferes with other mechanisms acting in parallel with Emp in restricting tracheal tube length (Laprise et al., 2010; Wang et al., 2006). Please also see our response to editors comment #1 above. We have considered reviewer's comment and we have commented in the text (see lines: 499-502 and 723-733).

The authors claim that Emp modulates the apical actin organization and that loss of Emp affects this F-actin organization. However, to analyse apical F-actin in DTs and compare control with emp mutants (Figure 6 A and B), the authors should present not just a median view of the tube lumen, but also show the cortex of the lumen in order to better analyze and visualize the F-actin bundles (as in Hannezo et al. 2015, Ozturk-Çolak 2016). In addition, this would give a better view of Emp localization in relation to the F-actin bundles. According to the model proposed Emp is clustered outside the F-actin-rich bundles. F-actin analysis in the cortex of the tubes together with Emp localization would allow for this analysis.

We have followed the reviewer's helpful suggestion. We imaged the tube cortex of wild type and *emp* mutant embryos over-expressing moe-GFP. We found that loss of *emp* affects this cortical F-actin organization. Actin bundles were diss-organized and their density was increased compared to wild type. We have added now image-projections showing this phenotype in comparison to the wild type (see Figure 6 suppl. 1E).

We agree. We have addressed the reviewer's suggestion now and provided a new figure showing the Emp localization in the cortex in relation to the actin bundles (Figure 2 —figure supplementary 3A-D). Briefly, we have stained wild type embryos over-expressing a GFPtagged version of the actin binding domain of moesin (*btl>moe-GFP*) with anti-Emp and antiGFP in tracheal cells. Analysis of *xz* or *yz* sections within Z-stacks of airy-scan confocal images in these embryos showed low colocalization (*r^2^* = 218, n = 5 embryos) and predominantly separate pattern of apical Emp and actin bundle domains (GFP). These results further support our proposed model that Emp is clustered outside the F-actin-rich bundles to promote the endocytosis and recycling of selected cargos along the longitudinal tube axis. See also our response in editor's comment #2.

In the discussion, the authors state that "First, emp activity is confined in apical "macro"-domains along the longitudinal tube axis by the transverse, DAAM-dependent actin filaments." This is really not shown in the manuscript. So, either the authors clarify Emp localization in "apical macro domains" and its relationship with DAAM-dependent actin filaments, or this should be removed.

Thank you for the comment. We would like to keep our statement in the discussion model because we have provided additional experimental data supporting the proposed complementary localization of Emp and F-actin bundles along the longitudinal tube axis (See editor's response, comments 2, Figure 2 —figure supplementary 3 (A-D) and below).

When discussing the model in figure 8, the authors say that "In the bundle-free membrane domains, Emp associates with Kst and establishes an endocytosis domain, where Serp mediated Emp clustering leads to internalization." This is not shown by their data and it should be made clear that this is a hypothesis and not a thesis.

We agree. In the new version, the sentence reads as follows:

"We hypothesize that in the bundle-free membrane domains, Emp associates with Kst and establishes an endocytosis domain, where Serp mediated Emp clustering leads to internalization" (lines: 805-808).

Ozturk-Çolak et al. (eLife 2016) presented a model whereby there is feedback between the chitinous aECM and the underlying cells. They reported that chitin downregulation increases the levels of p-Src in tracheal cells. There is a tight connection between chitin and the F-actin bundles, so I believe it is essential that the authors analyse the chitinous structure in the DT of Emp mutants. In addition, it would be very interesting (but not essential) to analyse if the chitinous luminal filament is also internalised via Emp.

Thank you for the comment. We stained for Gasp to visualize the chitinous matrix and analyzed the stainings with high-resolution confocal-microscopy (Airyscan). We were able to visualize the "ridges" of the apical extracellular matrix (taenidial folds) in wild type embryos and did not detect any disorganization in *emp* mutants (See the revised Figure 6C). However, we readily detected the disorganization of the actin bundles in the same type of parallel experiment with the same resolution (see Figure 6 suppl. 1E). Src42A phosphorylation levels has been proposed as a feedback mechanism of the aECM to the underlying cells to modulate proper cytoskeletal organization in airways. Our work is complementary to this mechanism and further argues that Emp organizes the apical cytoskeleton by its association with bH-Spectrin. Emp provides a first direct link between an ECM component, Serp, to the cytoskeleton, through βH-Spectrin. The previously proposed feedback mechanism where the ECM composition and physical properties additionally reinforce the actin bundle organization, is part of our model in the discussion.

Furthermore, the feedback model proposed by Ozturk-Çolak et al. could be very well integrated into this new model. The link between the aECM and the underlying cells and the modulation of the feedback could be sensed by Emp. This should be discussed.

We agree with the reviewer. We believe that our model complements the proposed model in (Ozturk-Colak et al., 2016). We have also added this statement in the discussion (lines: 820-827).

Reviewer #3 (Recommendations for the authors):One potentially insightful result is the phenotype of Serp-GFP overexpression, which caused a reduction of pSrc (Figure 7A-C) and elevation of membrane Emp (Figure 4A). Although down-regulation of Src and upregulation of Emp in other contexts are associated with defective tube elongation, btl>Serp-GFP caused tube over elongation (Figure 4G). This apparent contradiction should deserve highlighted, and a deeper investigation may open a better understanding of the tube elongation mechanism. One potential clue could be to look at the anisotropy of junctional Crumbs in the btl>Serp-GFP embryos.

Thanks for the comment and the suggestion. We agree that it is difficult to account for the tube over-elongation phenotype caused by Serp overexpression (see editor's-comment #1 and reviewer's responses above). We had shown before that Serp-GFP overexpression unexpectedly causes similar tube elongation as the double mutant of *serp verm* indicating that the levels of luminal Serp are important for its normal function. Nevertheless, we have followed the suggestion from the reviewer and found that *btl>Serp-GFP* overexpressing embryos showed decreased Crb signals along both longitudinal and transverse junctions, in contrast to emp mutants where Crb signals were only affected along the longitudinal junctions. This suggests that the "artificially" high levels of Serp may interfere with Crb levels and the general tensile properties of the junctions.

We iterate that tube elongation involves several mechanisms including SJ components, Crb retrograde transport (Dong et al., 2013; Olivares-Castineira and Llimargas, 2017), transcriptional control (Hemphälä et al., 2003; Robbins et al., 2014; Öztürk-Çolak et al., 2018). Future experiments are needed to elucidate the mechanism by which high Serp levels interferes with tube over-elongation.

**Author response image 2. sa2fig2:** (A) Confocal images showing the tracheal DT stained for Crb (yellow) in wild-type and btl>SerpFLembryos. (B) Plots showing the total fluorescence intensity of Crb in the Longitudinal and Transverse axis in wild-type and btl>SerpFLembryos.